# Training Over-parameterized Models with Non-decomposable Objectives

**Harikrishna Narasimhan**
Google Research, Mountain View
hnarasimhan@google.com

**Aditya Krishna Menon**
Google Research, New York
adityakmenon@google.com

## Abstract

Many modern machine learning applications come with complex and nuanced design goals such as minimizing the worst-case error, satisfying a given precision or recall target, or enforcing group-fairness constraints. Popular techniques for optimizing such non-decomposable objectives reduce the problem into a sequence of cost-sensitive learning tasks, each of which is then solved by re-weighting the training loss with example-specific costs. We point out that the standard approach of re-weighting the loss to incorporate label costs can produce unsatisfactory results when used to train over-parameterized models. As a remedy, we propose new cost-sensitive losses that extend the classical idea of logit adjustment to handle more general cost matrices. Our losses are calibrated, and can be further improved with *distilled* labels from a teacher model. Through experiments on benchmark image datasets, we showcase the effectiveness of our approach in training ResNet models with common robust and constrained optimization objectives.

## 1 Introduction

The misclassification error is the canonical performance measure in most treatments of classification problems [92]. While appealing in its simplicity, practical machine learning applications often come with more complex and nuanced design goals. For example, these may include minimizing the worst-case error across all classes [64], satisfying a given precision or recall target [106], enforcing minimal coverage for minority classes [31], or imposing group-fairness constraints [105]. Unlike the misclassification error, such objectives are *non-decomposable*, i.e., they cannot be expressed as a sum of losses over individual samples. This poses a non-trivial optimization challenge, which is typically addressed by reducing the problem into a sequence of cost-sensitive learning tasks [12, 25, 4, 16, 71]. Such reduction-based approaches have been successfully employed in many open-source libraries [3, 2, 1] to provide drop-in replacements for standard loss functions.

In this paper, we point out that the standard approach of re-weighting the training loss to incorporate label costs can produce unsatisfactory results when used with high capacity models, which are often trained to memorize the training data. Such *over-parameterized models* are frequently encountered in the use of modern neural networks, and have been the subject of considerable recent study [107, 73, 66]. As a remedy, we provide new calibrated losses for cost-sensitive learning that are better equipped at training over-parameterized models to optimize non-decomposable metrics, and demonstrate their effectiveness on benchmark image classification tasks. Our main contributions are as follows:

(i) we illustrate the pitfalls of using loss re-weighting in over-parameterized settings, particularly with diagonal class weights (Section 3).

(ii) we propose new logit-adjusted losses for cost-sensitive learning for both diagonal and non-diagonal gain matrices, and show that they are *calibrated* (Section 4).

(iii) we demonstrate that our losses provide significant gains over loss re-weighting in training ResNet models to optimize worst-case recall and to enforce coverage constraints (Section 5).

35th Conference on Neural Information Processing Systems (NeurIPS 2021).

(iv) we show that the proposed approach compares favorably to post-hoc correction strategies, and can be further improved by *distilling* scores from a teacher model (Section 6).

## 1.1 Related Work

We cover prior work on class-imbalanced learning, cost-sensitive learning, and complex metrics.

**Learning under class imbalance**. A common setting where metrics beyond the misclassification error have been studied is the problem of class imbalance [49, 11, 33]. Here, the label distribution $\mathbf{P}(y)$ is skewed, and one seeks to ensure that performance on rare classes do not overly suffer. To this end, common metrics include the average of per-class recalls (also known as the balanced accuracy) [7, 62], quadratic mean of per-class accuracies [51], and the F-score [54].

A panoply of different techniques have been explored for this problem, spurred in part by recent interest in the specific context of neural networks (where the problem is termed *long-tail learning*) [91, 9], with the focus largely on optimizing balanced accuracy. An exhaustive survey is beyond the scope of this paper (see, e.g,. [33, 39]), but we may roughly identify three main strands of work: data modification [48, 11, 93, 103, 108], loss modification [107, 18, 10, 88, 37, 79, 97, 63, 21, 45, 94], and prediction modification [28, 43, 76, 60, 13, 40, 110]. A related recent thread seeks to improve performance on rare *subgroups* [82, 83, 86], as a means to ensure model fairness [24].

Amongst loss modification techniques, two strategies are of particular relevance. The first is *re-weighting* techniques [100, 65, 18]. As has been noted [55, 96, 61] (and as we shall subsequently verify), these may have limited effect on training samples that are perfectly separable under a given model class (as is the case with overparameterized models). The second is *margin* techniques, which enforce a class-dependent margin in the loss [10, 88, 79, 63, 94]. This seeks to ensure that rare classes are separated with greater confidence, to account for the higher uncertainty in their decision boundary.

**Cost-sensitive learning**. Cost-sensitive learning is a classical extension of standard multiclass classification, wherein errors on different classes incur different costs [26, 23, 85]. Strategies for this problem largely mirror those for class imbalance, which can be seen as imposing a particular set of costs dependent on the label frequencies [58]. In particular, loss modification techniques based on asymmetric weights [57, 98, 104, 5, 19, 22, 85, 112] and margins [61, 44, 36, 45] have been explored, with the latter proving useful on separable problems. Amongst these, Lin et al. [57] handle a general cost matrix for multiclass problems, but require the class scores to sum to 0, a constraint that might be difficult to impose with neural networks. Khan et al. [44] propose a multiclass loss similar to the logit adjustment idea used in this paper, but only handle a specific type of cost matrix (e.g., in their setup, the default cost matrix is a matrix of all 1s). The standard multiclass loss of Crammer and Singer [17] can also be extended to handle a general cost matrix [90], but unfortunately is not calibrated [78].

**Complex metrics.** There has been much work on extending the class imbalance literature to handle more complex metrics and to impose data-dependent constraints. These methods can again be divided into loss modifications [38, 74, 41, 69, 42, 31, 25, 4, 16, 71, 27, 59, 50] and post-hoc prediction modifications [102, 68, 46, 70, 72, 32, 20, 67, 101, 89], and differ in how they decompose the problem into simpler cost-sensitive formulations (see [71] for a unified treatment of common reduction strategies). Most of these papers experiment with simple models, with the exception of Sanyal et al. [84], who use re-weighting strategies to train shallow DNNs, Kumar et al. [50], who train CNNs on binary-labeled problems, and Eban et al. [25], who train InceptionNets to optimize an AUC-based ranking metric. There have also been other attempts at training neural networks with ranking metrics [87, 35, 8, 77]. In contrast, our focus is on handling a general family of evaluation metrics popular in *multiclass* classification problems.

## 2 Non-decomposable Objectives

**Notations.** Let $[m] = \{1, \ldots, m\}$. Let $\Delta_m$ be the $(m-1)$-dimensional simplex with $m$ coordinates. Let $\mathbf{1}_m \in \mathbb{R}^m$ denote an all 1s vector of size $m$, and $\operatorname{diag}(u_1, \ldots, u_m)$ denote a $m \times m$ diagonal matrix with diagonal elements $u_1, \ldots, u_m$. For vectors $\mathbf{a}, \mathbf{b} \in \mathbb{R}^m$, $\mathbf{a}/\mathbf{b}$ denotes element-wise division. Let $\operatorname{softmax}_i(\mathbf{s}) = \frac{\exp(s_i)}{\sum_{j=1}^m \exp(s_j)}$ denote a softmax transformation of scores $\mathbf{s} \in \mathbb{R}^m$.

Consider a multiclass problem with an instance space $\mathcal{X} \subseteq \mathbb{R}^d$ and a label space $\mathcal{Y} = [m]$. Let $D$ denote the underlying data distribution over $\mathcal{X} \times [m]$, $D_{\mathcal{X}}$ denote the marginal distribution over the

---

**Algorithm 1** Reductions-based Algorithm for Maximizing Worst-case Recall (1)

---

**Inputs:** Training set $S$, Validation set $S^{\text{val}}$, Step-size $\omega \in \mathbb{R}_+$, Class priors $\boldsymbol{\pi}$, CS-loss $\ell_{\mathbf{G}}$
**Initialize:** Classifier $h^0$, Multipliers $\boldsymbol{\lambda}^0 \in \Delta_m$
**for** $t = 0$ to $T - 1$ **do**
   **Update** $\boldsymbol{\lambda}$**:**

$\qquad \lambda_i^{t+1} = \lambda_i^t \exp\left(-\omega \frac{\widehat{C}_{ii}[h^t]}{\pi_i}\right), \forall i$, where $\widehat{C}_{ij}[h] = \frac{1}{|S^{\text{val}}|} \sum_{(x,y) \in S^{\text{val}}} \mathbf{1}(y = i, h(x) = j)$

$\qquad \lambda_i^{t+1} = \frac{\lambda_i^{t+1}}{\sum_{j=1}^m \lambda_j^{t+1}}, \forall i$

$\qquad \mathbf{G} = \text{diag}(\lambda_1^{t+1}/\pi_1, \ldots, \lambda_m^{t+1}/\pi_m)$

   **Cost-sensitive Learning (CSL):**

$\qquad \mathbf{s}^{t+1} \in \text{argmin}_{\mathbf{s}} \frac{1}{|S|} \sum_{(x,y) \in S} \ell_{\mathbf{G}}(y, \mathbf{s}(x))$     // Replaced by few steps of SGD

$\qquad h^{t+1}(x) \in \text{argmax}_{i \in [m]} s_i^{t+1}(x), \forall x$

**end for**
**return** $h^T$

---

instances $\mathcal{X}$, and $\pi_i = \mathbf{P}(y = i)$ denote the class priors. We will use $p_i(x) = \mathbf{P}(y = i|x)$ to denote the conditional-class probability for instance $x$. Our goal is to learn a classifier $h : \mathcal{X} \to [m]$, and will measure its performance in terms of its confusion matrix $\mathbf{C}[h]$, where

$$C_{ij}[h] = \mathbf{E}_{(x,y) \sim D}\left[\mathbf{1}\left(y = i,\ h(x) = j\right)\right].$$

**Complex learning problems.** In the standard setup, one is often interested in maximizing the overall classification accuracy $\text{acc}[h] = \sum_i C_{ii}[h]$. However, in many practical settings, one may care about other metrics such as the recall for class $i$, $\text{rec}_i[h] = \frac{C_{ii}[h]}{\pi_i}$, the precision on class $i$, $\text{prec}_i[h] = \frac{C_{ii}[h]}{\sum_j C_{ji}}$, or the proportion of predictions made on class $i$, i.e. the coverage, $\text{cov}_i[h] = \sum_j C_{ji}$. Below are common examples of real-world design goals based on these metrics.

*Example 1* (Maximizing worst-case recall). In class-imbalanced settings, where the classifier tends to generalize poorly on the rare classes, we may wish to change the training objective to directly maximize the *minimum recall* across all classes [12]:

$$\max_h \min_{i \in [m]} \frac{C_{ii}[h]}{\pi_i}. \tag{1}$$

*Example 2* (Constraining per-class coverage). Another consequence of label imbalance could be that the proportion of predictions that the classifier makes for the tail classes is lower than the actual prevalence of that class. The following learning problem seeks to maximize the average recall while explicitly constraining the classifier's coverage for class $j$ to be at least 95% of its prior $\pi_j$ [16, 71]:

$$\max_h \frac{1}{m} \sum_{i=1}^m \frac{C_{ii}[h]}{\pi_i} \quad \text{s.t.} \quad \sum_{i=1}^m C_{ij}[h] \geq 0.95 \times \pi_j, \forall j \in [m]. \tag{2}$$

Other examples include constraints on recall and precision (see Table 1), the F-measure, and the area under the ROC and PR curves, as well as fairness constraints like equal opportunity, which are computed on group-specific versions of the confusion matrix [25, 71, 15, 32, 4].

**Reduction to cost-sensitive learning.** All the problems described above are *non-decomposable*, in the sense that, they cannot be written as minimization of a sum of errors on individual examples, and hence cannot be tackled using common off-the-shelf learning algorithms. The dominant approach for solving such problems is to formulate a sequence of cost-sensitive objectives, which do take the form of an average of training errors [70, 12, 4, 16]. For example, the robust learning problem in (1) can be equivalently re-written as the following saddle-point optimization problem:

$$\max_h \min_{\boldsymbol{\lambda} \in \Delta_m} \sum_{i=1}^m \lambda_i \frac{C_{ii}[h]}{\pi_i}, \tag{3}$$

where $\lambda_i$ is a multiplier for class $i$. One can then find a saddle-point for (3) by jointly minimizing the weighted objective over $\boldsymbol{\lambda} \in \Delta_m$ (using e.g. exponentiated-gradient updates) and maximizing

| | Problem | Gain Matrix | Losses |
|---|---|---|---|
| 1 | $\max_h \min_y \mathrm{rec}_y[h]$ | $\mathrm{diag}(\boldsymbol{\lambda}/\boldsymbol{\pi})$ | $\ell^{\mathrm{LA}}$ |
| 2 | $\max_h \mathrm{acc}[h]$ s.t. $\mathrm{rec}_y[h] \geq \tau, \forall y$ | $\mathrm{diag}(\mathbf{1}_m + \boldsymbol{\lambda}/\boldsymbol{\pi})$ | $\ell^{\mathrm{LA}}$ |
| 3 | $\max_h \mathrm{acc}_y[h]$ s.t. $\mathrm{prec}_y[h] \geq \tau, \forall y$ | $\mathrm{diag}(\mathbf{1}_m + \boldsymbol{\lambda}) - \tau \mathbf{1}_m \boldsymbol{\lambda}^\top$ | $\ell^{\mathrm{hyb}}, \ell^{\mathrm{SMS}}$ |
| 4 | $\max_h \sum_y \mathrm{rec}_y[h]$ s.t. $\mathrm{cov}_y[h] \geq 0.95\pi_y, \forall y$ | $\mathrm{diag}(\mathbf{1}_m/\boldsymbol{\pi}) + \mathbf{1}_m \boldsymbol{\lambda}^\top$ | $\ell^{\mathrm{hyb}}, \ell^{\mathrm{SMS}}$ |
| 5 | $\max_h \sum_y \mathrm{rec}_y[h]$ s.t. $\mathrm{bcov}_y[h] \geq 0.95\frac{1}{m}, \forall y$ | $\mathrm{diag}(\mathbf{1}_m/\boldsymbol{\pi}) + (\mathbf{1}_m/\boldsymbol{\pi})\boldsymbol{\lambda}^\top$ | $\ell^{\mathrm{hyb}}$ |

Table 1: Examples of complex learning problems, the associated gain matrices $\mathbf{G}$, and the proposed losses that are applicable to the setting. We use $\tau$ to denote the minimum allowed recall/precision, and $\mathrm{bcov}_y[h] = \sum_{i=1}^m \frac{1}{\pi_i} C_{iy}[h]$ is the balanced coverage metric we use in our experiments. Similar reductions to cost-sensitive learning can also be accomplished with other common evaluation metrics such as F-measure, G-mean, AUC-ROC, AUC-PR [25, 71], and fairness criteria such as equal opportunity and equalized odds [4, 89]. See Appendix A for details.

the objective over $h$. Notice that for a fixed $\boldsymbol{\lambda}$, the optimization over $h$ is a cost-sensitive (or a gain-weighted) learning problem:

$$\max_h \sum_{i,j} G_{ij} C_{ij}[h], \tag{4}$$

where $G_{ii} = \frac{\lambda_i}{\pi_i}$ and $G_{ij} = 0, \forall i \neq j$ are the rewards associated with predicting class $j$ when the true class is $i$. We will refer to $\mathbf{G} \in \mathbb{R}^{m \times m}$ as the "gain" matrix, which in this case is *diagonal*. In practice, the cost-sensitive learning problem in (4) is usually replaced with multiple steps of gradient descent, interleaved with the updates on $\boldsymbol{\lambda}$. Algorithm 1 provides an outline of this procedure.

Similar to (1), the constrained learning problem in (2) can be written as an equivalent (Lagrangian) min-max problem with Lagrange multipliers $\boldsymbol{\lambda} \in \mathbb{R}_+^m$:

$$\max_h \min_{\boldsymbol{\lambda} \in \mathbb{R}_+^m} \frac{1}{m} \sum_{i=1}^m \frac{1}{\pi_i} C_{ii}[h] + \sum_{j=1}^m \lambda_j \Big( \sum_{i=1}^m C_{ij}[h] - 0.95\pi_j \Big).$$

One can find a saddle-point for this problem by jointly minimizing the Lagrangian over $\boldsymbol{\lambda} \in \mathbb{R}_+^m$ (using e.g. gradient updates) and maximizing it over $h$, with the maximization over $h$ taking the form of a cost-sensitive learning problem. In this case, the gain matrix $\mathbf{G}$ is non-diagonal and is given by $G_{ii} = \frac{1}{m\pi_i} + \lambda_i, \forall i$ and $G_{ij} = \lambda_j, \forall i \neq j$. See Table 1 for the form of the gain matrix for other common constraints. In Appendix A, we provide an outline of this procedure (Algorithm 2), and also elaborate on how this reduction-based approach can be extended to optimize other common evaluation metrics such F-measure [25, 71].

**Cost-sensitive losses.** A standard approach for solving the cost-sensitive learning problem in (4) is to use a surrogate loss function $\ell : [m] \times \mathbb{R}^m \to \mathbb{R}_+$ that takes a label $y$ and a $m$-dimensional score $\mathbf{u} \in \mathbb{R}^m$, and outputs a real value $\ell(y, \mathbf{u})$. One would then *minimize* the expected loss $\mathcal{L}(\mathbf{s}) = \mathbf{E}_{x,y}[\ell(y, \mathbf{s}(x))]$ over a class of scoring function $\mathbf{s} : \mathcal{X} \to \mathbb{R}^m$ that map each instance to an $m$-dimensional score. The final classifier $h^*$ can then be obtained from the learned scoring function $\mathbf{s}^*$ by taking an argmax of its predicted scores, i.e. by constructing $h^*(x) \in \mathrm{argmax}_{i \in [m]} s_i^*(x)$.

In practice, we are provided a finite sample $S = \{(x_1, y_1), \ldots, (x_n, y_n)\}$ drawn from $D$, and will seek to minimize the average loss on $S$, given by $\widehat{\mathcal{L}}(\mathbf{s}) = \frac{1}{|S|} \sum_{(x,y) \in S} \ell(y, \mathbf{s}(x))$. We will also assume access to a held-out validation sample $S^{\mathrm{val}} = \{(x_1, y_1), \ldots, (x_{n^{\mathrm{val}}}, y_{n^{\mathrm{val}}})\}$.

One common sanity check for a "good" loss function is to confirm that it is *classification calibrated* (or Fisher consistent) for the learning problem of interest [6]. Loosely speaking, a surrogate loss $\ell$ is classification calibrated if the minimizers of the conditional risk $\mathbf{E}_{y|x}(\ell(y, \mathbf{s}(x)))$ over all scoring functions $\mathbf{s} : \mathcal{X} \to \mathbb{R}^m$ coincide with the Bayes-optimal prediction for $x$. For the cost-sensitive objective in (4), the Bayes-optimal classifier is given below [53].

**Proposition 1.** *The optimal classifier for* (4) *for a general gain matrix* $\mathbf{G} \in \mathbb{R}^{m \times m}$ *is of the form:*

$$h^*(x) \in \mathrm{argmax}_{y \in [m]} \sum_{i=1}^m G_{iy}\, p_i(x) = \mathrm{argmax}_{y \in [m]} (\mathbf{G}^\top \mathbf{p}(x))_y.$$

All proofs can be found in Appendix B.

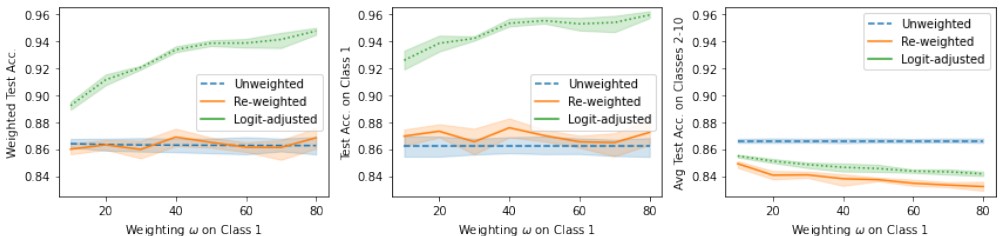

Figure 1: Training ResNet-56 on a subset of CIFAR-10 with an unweighted, re-weighted (cf. (5)) and logit-adjusted (cf. (7)) cross-entropy loss. All models achieve *near-zero training error*.

## 3 The Perils of Over-parameterization

We now point out the problems in applying the algorithms described previously to models that are over-parameterized, i.e., which *contain enough parameters to perfectly fit the labels in the training set.* We will focus particularly on the loss used for cost-sensitive learning steps in Algorithms 1–2.

**Limitations of loss re-weighting.** One of the most common loss function for a diagonal gain matrix $\mathbf{G}$ is a simple re-scaling of the standard cross-entropy loss with the diagonal weights:

$$\ell^{\mathrm{wt}}(y, \mathbf{s}) = -G_{y,y} \log \left( \frac{\exp(s_y)}{\sum_j \exp(s_j)} \right). \tag{5}$$

The following is a natural extension of this re-weighted loss to a general gain matrix $\mathbf{G}$, used, for example, in constrained optimization libraries [3], and also for mitigating label noise [75]:

$$\ell^{\mathrm{wt}}(y, \mathbf{s}) = -\sum_{i=1}^{m} G_{y,i} \log \left( \frac{\exp(s_i)}{\sum_j \exp(s_j)} \right). \tag{6}$$

While this weighted loss is calibrated for $\mathbf{G}$ (see, e.g., [75]), it is often inadequate when training an over-parameterized model on a *finite* sample. This is evident when $\mathbf{G}$ is a diagonal matrix: such a model will usually memorize the training labels and achieve *zero* training loss for every class, *irrespective* of what we choose for the outer weighting. In such separable settings, it is unclear how the outer weighting will impact the model's out-of-sample performance on different classes.

For clarity, we provide a simple illustration in Figure 1, where we use a re-weighted loss to train ResNet-56 models on 10000 images from CIFAR-10 with different diagonal gain matrices $\mathbf{G}$. We assign a weight of $\omega$ to the first class "airplane", and a weight of 1 on all other classes (i.e. set $G_{1,1} = \omega$ and $G_{y,y} = 1, \forall y \neq 1$), and plot the normalized weighted accuracy $\sum_y G_{yy} C_{yy}[h] / \sum_y G_{yy}$ on the test set as we vary $\omega$. Compared to an unweighted cross-entropy loss, the re-weighted loss does not produce a significant change to the test metric, whereas the logit-adjusted loss (which we will discuss in Section 4) yields substantially better values. Probing closer into the test accuracy for each class, we see that the re-weighted loss yields slightly better accuracies for class 1, but is significantly worse-off on the other classes, suggesting that re-weighting has the effect of excessively focusing on the class with the higher weight at the cost of the other classes. Sagawa et al. [83] also make a similar observation when using importance weights to improve worst-group generalization, while Cui et al. [18], Cao et al. [10] observe that up-weighting the minority classes can lead to unstable optimization.

When $\mathbf{G}$ is not diagonal, the minimum training loss may not be zero. Nonetheless, the following proposition sheds some light on the scores learned by a model that achieves minimum training loss.

**Proposition 2.** *Suppose the instances $x$ in $S$ are all unique. Let $\widehat{\mathcal{L}}^{\mathrm{wt}}(\mathbf{s}) = \frac{1}{|S|} \sum_{(x,y) \in S} \ell^{\mathrm{wt}}(y, \mathbf{s}(x))$ denote the average loss on training sample $S$ with $\ell^{\mathrm{wt}}$ defined as in (6) for a gain matrix $\mathbf{G}$. Then a scoring function $\widehat{\mathbf{s}}$ that achieves the minimum value of $\widehat{\mathcal{L}}^{\mathrm{wt}}(\mathbf{s})$ over all $\mathbf{s} : \mathcal{X} \to \mathbb{R}^m$ is of the form:* $\mathrm{softmax}_i(\widehat{\mathbf{s}}(x)) = \frac{G_{y,i}}{\sum_j G_{y,j}}, \forall (x,y) \in S.$

Notice that the model output is invariant to re-scaling of the rows in $\mathbf{G}$, i.e., one can multiply each row of the gain matrix $\mathbf{G}$ by a different scalar, and the values memorized by the model for each training example will remain unchanged. On the other hand, re-scaling a column of $\mathbf{G}$ does substantially change the score learned for each example. While this does not tell us about how the model would

behave on unseen new examples, our experience has been that loss re-weighting is usually effective when used to control the amount of out-of-sample predictions that a model makes for a certain class (by suitably scaling the column for that class in $\mathbf{G}$), but does not work well when used to emphasize greater accuracy on a particular class. Consequently, we find that loss re-weighting usually fares better with metrics like coverage that depend only on the model predictions and not on the true labels, than with metrics like recall which depend on both the predictions and labels.

**Existing remedies for better generalization.** The literature offers some remedy to improve model generalization with non-standard objectives. Sagawa et al. [83] improve performance on rare sub-groups by regularizing the losses based on the size of each group, but their solution applies to a specific learning problem. Other remedies such as smarter re-weighting strategies [56, 18] or deferred re-weighting schedules [10] have proven to work well in specialized imbalanced settings.

The work that most closely relates to our setting is Cotter et al. [14], who propose a simple modification to the algorithms discussed in Section 2, with the use of two datasets: a training sample $S$, and a held-out validation sample $S^{\text{val}}$. They suggest using the validation sample to perform updates on the multipliers and in turn the gain matrix $\mathbf{G}$, and the training sample to solve the resulting cost-sensitive learning problem in (4). The latter would typically involve optimizing the empirical risk on the training set $\frac{1}{|S|} \sum_{(x,y) \in S} \ell_{\mathbf{G}}(y, \mathbf{s}(x))$, for some cost-sensitive loss $\ell_{\mathbf{G}}$. The intuition here is that, even when a model achieves very low training error, the estimate of $\mathbf{G}$ will accurately reflect the model's performance on held-out examples. While this modification improves our estimate of $\mathbf{G}$, it only provides a partial solution for over-parameterized settings, where the model would still struggle to generalize well if $\ell_{\mathbf{G}}$ happens to be the simple re-weighted loss in (6).

## 4 Cost-sensitive Losses Based on Logit Adjustment

We present new cost-sensitive losses that seek to avoid the problems mentioned above. We build on recent work on *logit adjustment*, shown previously to be effective in long tail settings [79, 63, 94].

**Diagonal gain matrix.** When the gain matrix $\mathbf{G}$ is diagonal, Proposition 1 tells us that the Bayes-optimal classifier for the resulting weighted accuracy metric is of the form $h^*(x) \in \operatorname{argmax}_{y \in [m]} G_{yy} \, p_y(x)$, where $p_y(x) = \mathbf{P}(y|x)$. Intuitively, we would like the learned scoring function $\mathbf{s}(x)$ to mimic the Bayes-optimal scores $G_{yy} \, p_y(x)$ for each $x$. In particular, we would like the maximizer of $s_y(x)$ over labels $y$ to match the Bayes-optimal label for $x$.

One way to facilitate this is to adjust the scores $s_y(x)$ based on the diagonal weights $G_{yy}$, and to compute a loss on the adjusted scores. Specifically, we shift $s_y(x)$ to $\bar{s}_y(x) = s_y(x) - \log(G_{yy})$, and optimize the shifted scores so that their softmax transformation matches the class probabilities $\mathbf{p}(x)$. This would then encourage the original scorer $\mathbf{s}$ to be monotonic in the Bayes optimal scores:

$$\frac{\exp(\bar{s}_y(x))}{\sum_j \exp(\bar{s}_j(x))} = p_y(x) \iff \bar{s}_y(x) \propto \ln(p_y(x)) \iff s_y(x) \propto \ln(G_{yy} \, p_y(x)).$$

In practice, $\mathbf{p}(x)$ is not directly available to us, and so we employ the following logit-adjusted cross-entropy loss that implements this idea with labels $y$ drawn according to $\mathbf{p}(x)$:

$$\ell^{\text{LA}}(y, \mathbf{s}) = -\log\left(\frac{\exp(s_y - \log(G_{yy}))}{\sum_j \exp(s_j - \log(G_{jj}))}\right). \tag{7}$$

This loss is a simple generalization of the one analyzed in Ren et al. [79], Menon et al. [63], Wang et al. [94], in which the logits are adjusted based on the class priors to optimize the balanced error rate. Such approaches have historical precedent in the class imbalance literature [76, 111, 13].

**Proposition 3.** *The logit-adjusted loss $\ell^{\text{LA}}$ is calibrated for a diagonal gain matrix $\mathbf{G}$.*

Unlike the weighted loss in (5), changing the diagonal entries of $\mathbf{G}$ in (7) changes the *operating point* of the learned scoring function. In separable settings, this would mean that the diagonal weights will determine the operating point at which the model achieves zero training error, and tend to push the separator towards classes that have higher weights, thus helping improve model generalization. An alternate explanation posed by Menon et al. [63] re-writes the loss in a pair-wise form:

$$\ell^{\text{LA}}(y, \mathbf{s}) = \log\left(1 + \sum_{j \neq y} \exp\left(\delta_{yj} - (s_y - s_j)\right)\right), \tag{8}$$

where $\delta_{yj} = \log(G_{yy}/G_{jj})$ can be seen as the *relative margin* between class $y$ and class $j$. This tells us that the loss encourages a larger separation between classes that have different weights. In Appendix C, we elaborate on this margin interpretation and point out that this loss can in fact be seen as a soft-approximation to the more traditional margin-based loss of Crammer and Singer [17].

**General gain matrix.** When the gain matrix $\mathbf{G}$ is non-diagonal, a simple logit adjustment no longer works. In this case, we propose a hybrid approach that combines logit adjustment with an outer weighting. To this end, we prescribe factorizing the gain matrix into a product $\mathbf{G} = \mathbf{M}\mathbf{D}$ for some diagonal matrix $\mathbf{D} \in \mathbb{R}^{m \times m}$, and $\mathbf{M} = \mathbf{G}\mathbf{D}^{-1}$. The proposed loss then adjusts the logits to account for the diagonal entries of $\mathbf{D}$ and applies an outer weighting to account for $\mathbf{M}$:

$$\ell^{\mathrm{hyb}}(y, \mathbf{s}) \;=\; -\sum_{i=1}^{m} M_{yi} \log \left( \frac{\exp(s_i - \log(D_{ii}))}{\sum_j \exp(s_j - \log(D_{jj}))} \right). \tag{9}$$

In practice, $\mathbf{D}$ can be chosen to reflect the relative importance of the classes and include information such as the class priors, which cannot be effectively incorporated as part of the outer weighting. One simple choice could be $\mathbf{D} = \mathrm{diag}(1/\pi_1, \ldots, 1/\pi_m)$. Another intuitive choice could be to $\mathbf{D} = \mathrm{diag}(G_{11}, \ldots, G_{mm})$, so that the residual matrix $\mathbf{M} = \mathbf{G}\mathbf{D}^{-1}$ that serves as the outer weighting has 1s in all its diagonal entries, and thus equal weights on the diagonal loss terms.

**Proposition 4.** *For any diagonal matrix $\mathbf{D} \in \mathbb{R}^{m \times m}$ with $D_{yy} > 0, \forall y$, and $\mathbf{M} = \mathbf{G}\mathbf{D}^{-1}$, the hybrid loss $\ell^{\mathrm{hyb}}$ is calibrated for $\mathbf{G}$.*

In some special cases, we may be able to avoid the outer-weighting: e.g., when the gain matrix is the sum of a diagonal matrix and a matrix with equal rows, i.e. $\mathbf{G} = \mathrm{diag}(\boldsymbol{\alpha}) + \mathbf{1}\boldsymbol{\beta}^{\top}$ (cf. Table 1, rows 3 & 4), the Bayes-optimal classifier from Proposition 1 is of the form $h^*(x) = \mathrm{argmax}_{y \in [m]} \alpha_y p_y(x) + \beta_y$. The additive terms $\beta_y$ in the optimal classifier can then be encoded in the loss as a shift to the softmax output, giving us the following *softmax-shifted* loss: for constant $C > 0$,

$$\ell^{\mathrm{SMS}}(y, \mathbf{s}) = -\log \left( C \frac{\exp(s_y - \log(\alpha_y))}{\sum_j \exp(s_j - \log(\alpha_j))} - \beta_y/\alpha_y \right). \tag{10}$$

**Proposition 5.** $\ell^{\mathrm{SMS}}$ *is calibrated for the gain matrix* $\mathbf{G} = \mathrm{diag}(\boldsymbol{\alpha}) + \mathbf{1}\boldsymbol{\beta}^{\top}$ *when* $C = 1 + \sum_y \beta_y/\alpha_y$.

See Appendix D for a practical variant this loss that avoids a negative value within the log.

## 5 Experimental Comparison with Loss Re-weighting

We now showcase that the proposed losses provide substantial gains over loss re-weighting through experiments on three benchmark datasets: CIFAR-10, CIFAR-100 [47], and TinyImageNet [52, 80] (a subset of the ImageNet dataset with 200 classes). Similar to [18, 10, 63], we use long-tail versions of these datasets by downsampling examples with an exponential decay in the per-class sizes. We set the imbalance ratio $\frac{\max_i \mathbf{P}(y=i)}{\min_i \mathbf{P}(y=i)}$ to 100 for CIFAR-10 and CIFAR-100, and to 83 for TinyImageNet (the slightly smaller ratio is to ensure that the smallest class is of a reasonable size). In each case, we use a balanced validation sample of 5000 held-out images, and a balanced test set of the same size.

We trained ResNet-56 models on CIFAR-10 and CIFAR-100, and ResNet-18 models on TinyImageNet, using SGD with momentum. We provide details about our hyper-parameters choices in Appendix E. On CIFAR-10 and CIFAR-100, the ResNet-56 architectures have sufficient parameters to perfectly fit the training labels, as evident from the training set metrics reported in Appendix E.1.[1]

**Cost-sensitive learning and baselines.** We consider two learning objectives: maximizing worst-case recall, and maximizing average recall subject to coverage constraints. As outlined in Algorithms 1–2, we employ the "two dataset" approach of Cotter et al. [14] (discussed in Section 3) to solve these problems, with the validation set used for updates on the gain matrix $\mathbf{G}$, and the training set used for the cost-sensitive learning (CSL) step. Since our goal here is to demonstrate that the the proposed losses fair better at solving the CSL step than loss re-weighting, we use a large validation sample to get good estimates of the gain matrix $\mathbf{G}$. For the CIFAR datasets, we perform 32 SGD steps on

---

[1]Code will be made available at: `https://github.com/google-research/google-research/tree/master/non_decomp`

| Method | CIFAR-10-LT | | CIFAR-100-LT | | TinyImgNet-LT | |
|---|---|---|---|---|---|---|
| | Avg Rec | Min Rec | Avg Rec | Min HT Rec | Avg Rec | Min HT Rec |
| ERM | 0.750 | 0.563 | 0.441 | 0.082 | 0.321 | 0.009 |
| Balanced [63] | 0.790 | 0.664 | 0.478 | 0.210 | 0.353 | 0.086 |
| Equalized [88] | 0.754 | 0.541 | 0.461 | 0.147 | 0.325 | 0.033 |
| Adaptive [10] | 0.756 | 0.584 | 0.450 | 0.116 | 0.323 | 0.005 |
| CSL [Re-weighted] | 0.739 | 0.619 | 0.424 | 0.089 | 0.300 | 0.094 |
| CSL [Logit-adjusted] | 0.786 | 0.731 | 0.462 | 0.407 | 0.335 | 0.295 |

Table 2: Results of maximizing worst-case recall on CIFAR-10 and the minimum of the head and tail recalls on CIFAR-100 and Tiny-ImageNet. The last two rows contain results from Algorithm 1 with different losses in the CSL step. All metrics are evaluated on the test set and averaged over 5 independent trials. The highest entry in each column is shaded. The proposed logit-adjusted loss yields significantly better worst-case recall than all the baselines.

| Method | CIFAR-10-LT | | CIFAR-100-LT | | TinyImgNet-LT | |
|---|---|---|---|---|---|---|
| | Avg Rec | Min Cov (tgt: 0.095) | Avg Rec | Min HT Cov (tgt: 0.010) | Avg Rec | Min HT Cov (tgt: 0.005) |
| ERM | 0.750 | 0.058 | 0.441 | 0.001 | 0.321 | 0.000 |
| Balanced [63] | 0.790 | 0.076 | 0.478 | 0.005 | 0.353 | 0.001 |
| Equalized [88] | 0.754 | 0.056 | 0.461 | 0.003 | 0.325 | 0.000 |
| Adaptive [10] | 0.756 | 0.060 | 0.450 | 0.002 | 0.323 | 0.000 |
| CSL [Re-weighted] | 0.750 | 0.087 | 0.411 | 0.010 | 0.295 | 0.002 |
| CSL [Hybrid A] | 0.750 | 0.088 | 0.461 | 0.010 | 0.347 | 0.005 |
| CSL [Hybrid B] | 0.755 | 0.087 | 0.464 | 0.010 | 0.353 | 0.004 |

Table 3: Results of maximizing average recall, while constraining the coverage for each class to be at least 95% of $\frac{1}{m}$ on CIFAR-10, and constraining the head and tail coverages to be at least 95% of $\frac{1}{m}$ on CIFAR-100 and TinyImageNet. The models are evaluated on a balanced test set and hence we set the coverage target to $0.95\frac{1}{m}$. The last three rows contain results from Algorithm 2 (in the appendix) with different losses in the CSL step. All metrics and targets are rounded off to 3 decimal places, and averaged over 5 independent trials. The highest entry and those comparable to it (within 0.001) are shaded. The maximum recall among rows where the coverage is closest to the target (or within 0.001 of the closest entry) is underlined. The proposed hybrid losses yield coverage values $\geq$ target, while achieving a higher average recall than the re-weighted loss.

the cost-sensitive loss for every update on $\mathbf{G}$, and for TinyImageNet, we perform 100 SGD steps for every update on $\mathbf{G}$. In each case, we compare the results with empirical risk minimization (ERM) with the standard cross-entropy loss, and as representative methods that seek to maximize the average per-class recall (i.e. balanced accuracy), we include the balanced logit-adjusted loss of Menon et al. [63] in which the adjustments are based on the class priors alone (Balanced), the equalized loss of Tan et al. [88] (Equalized), and the adaptive margin loss of Cao et al. [10] (Adaptive).

**Maximizing worst-case recall.** In our first task, we maximize the worst-case recall over all 10 classes on CIFAR-10 (cf. (1)). Given the larger number of classes, this can be a very restrictive goal for CIFAR-100 and TinyImageNet, leading to poor overall performance. So for these datasets, we will consider a simpler goal, where we measure the average recall on the bottom 10% of the smallest-sized classes (the "tail" recall), and the average recall on the top 90% of the classes (the "head" recall), and seek to maximize the minimum of the head and tail recalls:

$$\max_h \min \left\{ \frac{1}{|\mathcal{H}|} \sum_{y \in \mathcal{H}} \text{rec}_y[h], \ \frac{1}{|\mathcal{T}|} \sum_{y \in \mathcal{T}} \text{rec}_y[h] \right\},$$

where $\mathcal{H} \subset [m]$ and $\mathcal{T} \subset [m]$ denote the set of head and tail labels respectively. The gain matrix $\mathbf{G}$ for these problems is diagonal. In Table 2, we present results of maximizing the worst-case recall metrics using Algorithm 1, with both the re-weighted loss in (5) and the proposed logit-adjusted loss for a diagonal $\mathbf{G}$ in (7) used in the CSL step. Compared to loss re-weighting, the prescribed loss provides huge gains in the worst-case recalls. In fact, on the CIFAR datasets, loss re-weighting fairs significantly worse than using the simple logit adjustment with class priors prescribed by Menon et al. [63], which performs the best on the average recall metric that it seeks to maximize. Our approach improves substantially over this baseline on the worst-case recall, at the cost of a moderate decrease in the average recall.

| Method | C10-LT Per-class Recall | | C100-LT Per-class Recall | | C100-LT Head-Tail Recall | | TIN-LT Head-Tail Recall | |
|---|---|---|---|---|---|---|---|---|
| | Avg | Min | Avg | Min | Avg | Min | Avg | Min |
| ERM | 0.750 | 0.563 | 0.434 | 0.000 | 0.441 | 0.082 | 0.321 | 0.009 |
| ERM + PS | 0.766 | 0.711 | 0.279 | 0.103 | 0.459 | 0.454 | 0.319 | 0.303 |
| CSL [Logit-adjusted] | 0.786 | 0.731 | 0.389 | 0.107 | 0.462 | 0.407 | 0.335 | 0.295 |
| CSL [Logit-adjusted] + PS | 0.766 | 0.730 | 0.298 | 0.068 | 0.456 | 0.447 | 0.328 | 0.315 |

Table 4: Improvements with post-shifting: Results (on the test set) of maximizing the minimum recall over *all classes* (col 1–2) and over just the head and tail classes (col 3–4). The results are averaged over 5 independent trials. The highest entry in each column is shaded. The proposed approach, or its variant (CSL [Logit-adjusted] + PS), where the trained model is modified with a post-hoc adjustment, compares favorably to ERM + PS. Our proposal often yields better average recall.

| Method | CIFAR-10-LT | | CIFAR-100-LT | |
|---|---|---|---|---|
| | Avg Rec | Min Rec | Avg Rec | Min HT Rec |
| Distilled ERM | 0.766 | 0.600 | 0.456 | 0.057 |
| Distilled Balanced [63] | 0.812 | 0.709 | 0.508 | 0.230 |
| Distilled CSL [Re-weighted] | 0.771 | 0.735 | 0.481 | 0.478 |
| Distilled CSL [Logit-adjusted] | 0.777 | 0.737 | 0.475 | 0.473 |

Table 5: Improvements with self-distillation: Performance of student model (on the test set) in maximizing worst-case recall on CIFAR-10 and the minimum of the head and tail recalls on CIFAR-100. Both the teacher and student use a ResNet-56 architecture. The results are averaged over 5 independent trials. The highest entry in each column is shaded.

**Constraining coverage.** The next task we consider for CIFAR-10 seeks to ensure that when evaluated on a balanced dataset, the model makes the same proportion of predictions for each class. This leads us to the optimization problem shown in row 5 of Table 1, where we wish to maximize the average recall, constraining the model's "balanced coverage" on each class $\text{bcov}_y[h] = \sum_{i=1}^{m} \frac{1}{\pi_i} C_{iy}[h]$ to be at least 95% of $\frac{1}{m}$, where $m$ is the number of classes. Because the validation and test samples are already balanced, the model's coverage on these datasets is the same as its balanced coverage. For CIFAR-100 and TinyImageNet, we consider the simpler goal of maximizing the average recall over all classes, with constraints on the model's average coverage over the head labels, and its average coverage over the tail labels to be both at least 95% of $\frac{1}{m}$:

$$\max_h \frac{1}{m} \sum_{y \in [m]} \text{rec}_y[h] \quad \text{s.t.} \quad \frac{1}{|\mathcal{H}|} \sum_{y \in \mathcal{H}} \text{bcov}_y[h] \geq 0.95 \frac{1}{m}, \quad \frac{1}{|\mathcal{T}|} \sum_{y \in \mathcal{T}} \text{bcov}_y[h] \geq 0.95 \frac{1}{m}.$$

The gain matrix $\mathbf{G}$ for these problems (as shown in Table 1) is non-diagonal, and does not have a special structure that we can exploit. We will therefore use the hybrid loss function that we provide in (9) for a general $\mathbf{G}$, and try out both variants suggested: with the diagonal matrix $\mathbf{D} = \text{diag}(1/\pi_1, \ldots, 1/\pi_m)$ (variant "A"), and with $\mathbf{D} = \text{diag}(G_{11}, \ldots, G_{mm})$ (variant "B"). In Table 3, we present the results of applying Algorithm 2 (in the appendix) to this task, with both the re-weighted loss and the proposed losses used in the CSL step. In this case, loss re-weighting is able to match the coverage target as closely as the proposed losses on CIFAR-10 and CIFAR-100, but fares poorly on the average recall on two of the three datasets. The proposed losses are often considerably better on this metric, while yielding coverage values that are closest to the target. Between the two hybrid variants, there isn't a clear winner, although variant "B" seems to have a slight edge. The reason we are not able to match the target in CIFAR-10 as closely as we do in the other two datasets is because of the larger number of constraints (10) that need to be satisfied (we only have 2 constraints to be satisfied for CIFAR-100 and TinyImageNet).

# 6 Improvements with Post-shifting and Distillation

Having demonstrated that the proposed losses can provide significant gains over loss re-weighting, we explore ways to further improve their performance.

**Does post-shifting provide further gains?** Post-hoc correction strategies have generally shown to be very effective in optimizing evaluation metrics [68, 46, 101, 89, 63], often matching the performance of more direct methods that modify the training loss. They are implemented in two steps: (i) train a

base scoring model $\mathbf{s} : \mathcal{X} \to \mathbb{R}^m$ using ERM, (ii) construct a classifier that estimates the Bayes-optimal label for a given $x$ by applying a gain matrix $\mathbf{G} \in \mathbb{R}^{m \times m}$ to the predicted probabilities:

$$h(x) \in \operatorname{argmax}_{y \in [m]} \sum_{i=1}^{m} G_{iy} \eta_i(x), \quad \text{where} \quad \boldsymbol{\eta}(x) = \operatorname{softmax}(\mathbf{s}(x)).$$

The coefficients $\mathbf{G}$ are usually chosen to maximize the given evaluation metric on a held-out validation set, using either a simple grid search (when the number of classes is small) or more sophisticated optimization tools [70, 89]. Table 4 shows the results of post-shifting the ERM-trained model for the tasking of maximizing worst-case recall (see Appendix E for details of how we fit the post-shift coefficients). While post-shifting provides huge improvements over ERM, our proposed approach of employing an iterative training procedure (Algorithm 1) with a modified logit-adjusted loss for the CSL step does often have a slight edge over the simpler post-shifting approach on the worst-case recall. Moreover, our proposal often yields a higher average recall, e.g., on the more difficult problem of maximizing the worst-case recall over *all 100 classes* in CIFAR-100-LT. Interestingly, applying the post-hoc adjustment procedure to a model trained using our iterative approach (CSL [Logit-adjusted] + PS) does not always yield further gains. For example, while maximizing the worst-case recall over all 100 classes on CIFAR-100-LT, the resulting classifier from this hybrid approach overfits to validation sample and is worse off on the test sample.

**Improvements with distillation.** Another technique that has proven effective in boosting the performance of neural networks is *knowledge distillation*, wherein soft predictions from a "teacher" model $\mathbf{p}^t : \mathcal{X} \to \Delta_m$ are used as labels to train a "student" model [34, 81, 29, 99, 109]. All the losses discussed so far can be easily applied to a distillation setup, where the training labels can be replaced with an expectation over the teacher's label distribution: $\mathbf{E}_{x \sim D_{\mathcal{X}}} \left[ \sum_{y=1}^{m} p_y^t(x) \ell(y, \mathbf{s}(x)) \right]$. In applying these losses for cost-sensitive learning, it is important that the class priors $\pi_y$ used to construct the gain matrices (see Table 1) be replaced with the teacher's prior distribution $\pi_y^t = \frac{1}{|S|} \sum_{(x,y) \in S} p_y^t(x)$. This is particularly important when the teacher is trained on a dataset with a different prior distribution, or its outputs are re-calibrated to yield soft predictions. We make use of the teacher's labels only in the training sample, and retain the original one-hot labels in the validation sample.

We re-run the worst-case recall experiments on CIFAR-10 and CIFAR-100 from Section 5 with distillation, and compare a distilled version of our approach with the distilled ERM, a distilled version of the class-prior-adjusted loss of Menon et al. [63] (which performed the best in Table 2 among all methods that sought to maximize average recall), and a distilled version of loss re-weighting. In each case, we employ *self-distillation* and use the same ResNet-56 architecture for both the teacher and the student. As shown in Table 5, distillation provides improvements across the board. Interestingly, loss re-weighting is quite competitive in this case, and has a slight edge on CIFAR-100. We attribute this surprising improvement in performance to the teacher prior distribution $\pi_y^t$ being significantly more balanced than the original class priors, and as a result, the costs $G_{yy}$ in the optimization of the worst-case recall in Algorithm 1 being less drastically different across classes.

# 7 Conclusions

We have proposed new cost-sensitive losses based on logit adjustment for training over-parameterized models with non-decomposable metrics. Our losses serve as replacements for and provide significant gains over standard loss re-weighting strategies used as a part of the iterative reduction-based approach to optimize such metrics. We show that our proposal compares favorably to the common post-shifting baseline, and can be further improved with distillation. While we considered two representative tasks in our experiments (maximizing worst-case recall and constraining coverage), the reductions-based approach that we employ and the losses that we propose apply to general evaluation metrics such as the F-measure, the AUC-PR, and common fairness goals [25, 4, 67].

A limitation of our losses is that while they are calibrated, these guarantees only hold in the large sample limit. In the future, we wish to probe further into why our loses improve generalization even in finite sample settings, and to develop a more formal understanding of their margin properties. We would also like to further investigate into why post-hoc adjustment approaches work well in practice. More broadly, when applied to objectives with fairness constraints, the techniques presented here could be used to improve performance of complex neural networks on under-represented samples. Carefully studying their empirical behaviour in such settings, to ensure they do not introduce unforeseen additional biases, is an important direction for future study.

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
