# Supplementary Material for "Training Over-parameterized Models with Non-decomposable Objectives"

---

**Algorithm 2** Reductions-based Algorithm for Constraining Coverage (2)

---

1: Inputs: Training set $S$, Validation set $S^{\text{val}}$, Step-size $\omega \in \mathbb{R}_+$, Class priors $\boldsymbol{\pi}$, CS-loss $\ell_{\mathbf{G}}$
2: Initialize: Classifier $h^0$, Multipliers $\boldsymbol{\lambda}^0 \in \mathbb{R}_+^m$
3: **for** $t = 0$ to $T - 1$ **do**
4:     **Update $\boldsymbol{\lambda}$:**
5:       $\lambda_i^{t+1} = \lambda_i^t - \omega\big(\sum_{j=1}^m \widehat{C}_{ji}[h^t] - 0.95\pi_i\big), \forall i$
                          where $\widehat{C}_{ij}[h] = \frac{1}{|S^{\text{val}}|} \sum_{(x,y) \in S^{\text{val}}} \mathbf{1}(y = i, h(x) = j)$
6:       $\lambda_i^{t+1} = \max\{0, \lambda_i^{t+1}\}, \forall i$    // Projection to $\mathbb{R}_+$
7:       $G_{ij} = \frac{1}{m\pi_i}\mathbf{1}(i = j) + \lambda_j^{t+1}, \forall i, j$
8:     **Cost-sensitive Learning (CSL):**
9:       $\mathbf{s}^{t+1} \in \text{argmin}_{\mathbf{s}} \frac{1}{|S|} \sum_{(x,y) \in S} \ell_{\mathbf{G}}(y, \mathbf{s}(x))$   // Replaced by few steps of SGD
10:       $h^{t+1}(x) \in \text{argmax}_{i \in [m]} s_i^{t+1}(x), \forall x$
11: **end for**
12: **return** $h^T$

---

## A    Algorithms for Robust and Constrained Learning

Recall that the algorithms discussed in Section 2 have two intuitive steps: (i) update the multipliers $\boldsymbol{\lambda}$ based on the current classifier's performance and construct a gain matrix $\mathbf{G}$; (ii) train a new classifier by optimizing a cost-sensitive loss $\ell_{\mathbf{G}}$ for $\mathbf{G}$. Algorithm 1 in the main text outlined this procedure for the problem of maximizing the worst-case recall in (1), and Algorithm 2 provided here outlines this procedure for the problem of maximizing the average recall subject to coverage constraints in (2). These algorithms additionally incorporate the "two dataset" trick suggested by Cotter et al. [14] for better generalization, wherein the updates on $\boldsymbol{\lambda}$ are performed using a held-out validation set $S^{\text{val}}$, and the minimization of the resulting cost-sensitive loss is performed using the training set $S$.

In Algorithm 1, we seek to find a saddle-point for the max-min problem for (1). For this, we jointly minimize the weighted objective over $\boldsymbol{\lambda} \in \Delta_m$ using exponentiated-gradient descent and maximize the objective over $h$. The latter can be equivalently formulated as the minimization of a cost-sensitive loss $\ell_{\mathbf{G}}$ with $\mathbf{G} = \text{diag}(\lambda_1/\pi_1, \ldots, \lambda_m/\pi_m)$. In Algorithm 2, we seek to find a saddle-point for the Lagrangian max-min problem for (2). In this case, we jointly minimize the Lagrangian over $\boldsymbol{\lambda} \in \Delta_m$ using projected gradient descent and maximize the Lagrangian over $h$. The latter is equivalent to minimizing a cost-sensitive loss $\ell_{\mathbf{G}}$ with $\mathbf{G} = \text{diag}\big(\frac{\mathbf{1}_m}{m\boldsymbol{\pi}}\big) + \mathbf{1}_m \boldsymbol{\lambda}^\top$. In our experiments, the class priors $\boldsymbol{\pi}$ were estimated from the training sample.

The cost-sensitive learning steps optimizes a scoring function $\mathbf{s} : \mathcal{X} \rightarrow \mathbb{R}^m$ over a class of scoring models, and constructs a classifier $h(x) \in \text{argmax}_{i \in [m]} s_i(x)$ from the learned scoring function. In practice, we do not perform a full optimization for this step, and instead perform a few steps of stochastic gradient descent (SGD) on the loss $\ell_{\mathbf{G}}$, warm-starting each time from the scoring function from the previous iteration.

See Chen et al. [12], Cotter et al. [16] for theoretical guarantees for the learned classifier, which usually require the algorithms to output a stochastic classifier that averages over the individual iterates $h^1, \ldots, h^T$. Since a stochastic classifier can be difficult to deploy [15], in practice, for the problems we consider, we find it sufficient to simply output the last iterate $h^T$.

### A.1    Extension to General Metrics

The reduction-based iterative approach outlined in Algorithms 1–2 extend to general non-decomposable metrics such as the F-score (by introducing auxiliary variables), as well as, AUC-based metrics (via a Riemann approximation to the integral). Eban et al. [25] provide details of how the optimization of these metrics can be posed as constrained optimization problems, and in turn reduced

to a sequence of cost-sensitive learning tasks. More generally, using the reduction techniques from Narasimhan et al. [71], any learning problem of the following form can be reduced to cost-sensitive learning problems, and thus tackled by the proposed approach:

$$\max_h \psi(\mathbf{C}[h]) \ \text{ s.t. } \ \phi_k(\mathbf{C}[h]) \leq \mathbf{0}, \ k = 1, \ldots, K,$$

where $\psi$ and $\phi_k$s are convex (or fractional-linear) functions of the confusion matrix. Their reduction techniques are generic and only require to be able to compute gradients for $\psi$ and $\phi_k$s.

Similar to the formulation in Section 2, the $G$-matrix for general metrics of the above form can be constructed by writing out the Lagrangian for the problem, and using the Lagrange multipliers to determine the weights on individual classes. Narasimhan et al. [71] provide details of the Lagrangian primal-dual optimization for different metrics. For example, to maximize the F-score: $\frac{2\text{TP}[h]}{2\text{TP}[h]+\text{FP}[h]+\text{FN}[h]}$, one could introduce slack variables $\xi_1$ and $\xi_2$ for the numerator and the denominator, and reformulate the problem as:

$$\max_{h,\xi_1,\xi_2 \in [0,1]} \frac{\xi_1}{\xi_2} \ \ \text{s.t.} \ \ \xi_1 \leq 2\text{TP}[h], \ \ \xi_2 \geq 2\text{TP}[h] + \text{FP}[h] + \text{FN}[h].$$

The Lagrange multipliers $\lambda_1$ and $\lambda_2$ for the constraints can then be used to derive weights on TP, FP and FN (i.e., the $G$-matrix).

## B   Proofs

We will find the following standard result to be useful in our proofs. Since the negative log is a *strictly proper*, in the sense of Gneiting and Raftery [30], Williamson et al. [95], we have that:

**Lemma 6** (Gneiting and Raftery [30], Williamson et al. [95]). *For any distribution $\mathbf{u} \in \Delta_m$, the minimizer of the expected risk*

$$\mathbf{E}_{y \sim \mathbf{u}}\left[-\log(v_y)\right] \ = \ -\sum_{i=1}^m u_i \log(v_i)$$

*over all distributions $\mathbf{v} \in \Delta_m$ is* unique *and achieved at* $\mathbf{v} = \mathbf{u}$.

### B.1   Proof of Proposition 1

*Proof.* We reproduce the proof from Narasimhan et al. [70]. Expanding the weighted accuracy in (4),

$$\sum_{i,j} G_{ij}\, C_{ij}[h] = \mathbf{E}_{x,y}\Big[\sum_{i,j} G_{ij}\, \mathbf{1}(y=i, h(x)=j)\Big] \ = \ \mathbf{E}_{x,y}\Big[\sum_j G_{yj}\, \mathbf{1}(h(x)=j)\Big]$$

$$= \mathbf{E}_x\Big[\mathbf{E}_{y|x}\Big[\sum_j G_{yj}\, \mathbf{1}(h(x)=j)\Big]\Big] \ = \ \mathbf{E}_x\Big[\sum_{i,j} p_i(x)\, G_{ij}\mathbf{1}(h(x)=j)\Big].$$

To compute the Bayes-optimal classifier for (4), it suffices to maximize the above objective point-wise, and to predict for each $x$, the label which maximizes the term within the expectation:

$$h^*(x) \in \text{argmax}_{j \in [m]} \sum_i p_i(x)\, G_{ij} \ = \ \text{argmax}_{j \in [m]} (\mathbf{G}^\top \mathbf{p}(x))_j,$$

as desired. $\qquad\square$

### B.2   Proof of Proposition 2

*Proof.* Given that the training instances $x$ in $S$ are unique, the average training loss $\widehat{\mathcal{L}}^{\text{wt}}(\mathbf{s}) = \frac{1}{|S|}\sum_{(x,y) \in S} \ell^{\text{wt}}(y, \mathbf{s}(x))$ is minimized by a scoring function $\mathbf{s}$ that yields the minimum loss $\ell^{\text{wt}}(y, \mathbf{s}(x))$ for each $(x,y) \in S$. For a fixed $(x,y) \in S$, the loss can be expanded as:

$$\ell^{\text{wt}}(y, \mathbf{s}(x)) \ = \ -\sum_{i=1}^m G_{y,i} \log\left(\frac{\exp(s_i(x))}{\sum_j \exp(s_j(x))}\right) \ = \ -C_y \sum_{i=1}^m \frac{G_{y,i}}{\sum_j G_{y,j}} \log\left(\frac{\exp(s_i(x))}{\sum_j \exp(s_j(x))}\right),$$

where $C_y = \sum_j G_{y,j}$ can be treated as a constant for a fixed $y$. We then have from Lemma 6, that any scoring function $\widehat{\mathbf{s}}$ that minimizes $\widehat{\mathcal{L}}^{\text{wt}}(\mathbf{s})$ evaluates to $\frac{\exp(\widehat{s}_i(x))}{\sum_j \exp(\widehat{s}_j(x))} = \frac{G_{y,i}}{\sum_j G_{y,j}}$, $\forall i \in [m]$ on the examples $(x,y)$ in $S$. $\qquad\square$

## B.3 Proof of Propositions 3–4

We provide a proof for Proposition 4. The proof of Proposition 3 follows by setting $\mathbf{D} = \mathbf{G}$ in the hybrid loss in (9).

*Proof of Proposition 4.* We wish to show that for any fixed $x$, the scoring function $\mathbf{s}^* : \mathcal{X} \to \mathbb{R}^m$ that minimizes the expected loss $\mathbf{E}_{(x,y) \sim D} \left[ \ell^{\mathrm{hyb}}(y, \mathbf{s}(x)) \right]$ recovers the Bayes-optimal classifier for $\mathbf{G}$. Appealing to Proposition 1, this would require us to show that for any $x$:

$$\mathrm{argmax}_{y \in [m]} s_y^*(x) \subseteq \mathrm{argmax}_{y \in [m]} (\mathbf{G}^\top \mathbf{p}(x))_y. \tag{11}$$

To this end, we first re-write the expected loss in terms of a conditional risk:

$$\mathbf{E}_{(x,y) \sim D} \left[ \ell^{\mathrm{hyb}}(y, \mathbf{s}(x)) \right] = \mathbf{E}_{x \sim D_{\mathcal{X}}} \left[ \mathbf{E}_{y \sim \mathbf{p}(x)} \left[ \ell^{\mathrm{hyb}}(y, \mathbf{s}(x)) \right] \right].$$

The optimal scoring function $\mathbf{s}^*(x)$ therefore minimizes the conditional risk $\mathbf{E}_{y \sim \mathbf{p}(x)} \left[ \ell^{\mathrm{hyb}}(y, \mathbf{s}(x)) \right]$ for each $x$. Expanding the conditional risk for a fixed $x$, we have:

$$
\begin{aligned}
\mathbf{E}_{y \sim \mathbf{p}(x)} \left[ \ell^{\mathrm{hyb}}(y, \mathbf{s}(x)) \right] &= \sum_{y=1}^m p_y(x) \, \ell^{\mathrm{hyb}}(y, \mathbf{s}(x)) \\
&= -\sum_{i=1}^m \sum_{y=1}^m M_{yi} \, p_y(x) \log \left( \frac{\exp(s_i(x) - \log(D_{ii}))}{\sum_j \exp(s_j(x) - \log(D_{jj}))} \right) \\
&= -\sum_{i=1}^m (\mathbf{M}^\top \mathbf{p}(x))_i \log \left( \frac{\exp(s_i(x) - \log(D_{ii}))}{\sum_j \exp(s_j(x) - \log(D_{jj}))} \right) \\
&= -C \sum_{i=1}^m \frac{(\mathbf{M}^\top \mathbf{p}(x))_i}{\sum_j (\mathbf{M}^\top \mathbf{p}(x))_j} \log \left( \frac{\exp(s_i(x) - \log(D_{ii}))}{\sum_j \exp(s_j(x) - \log(D_{jj}))} \right),
\end{aligned}
$$

where $C = \sum_j (\mathbf{M}^\top \mathbf{p}(x))_j$ can be treated as a constant for a fixed $x$. Appealing to Lemma 6, we then have that for any fixed $x$, because $\mathbf{s}^*(x)$ minimizes the conditional risk,

$$\frac{\exp(s_i^*(x) - \log(D_{ii}))}{\sum_j \exp(s_j^*(x) - \log(D_{jj}))} = \frac{(\mathbf{M}^\top \mathbf{p}(x))_i}{\sum_j (\mathbf{M}^\top \mathbf{p}(x))_j}, \forall i \in [m].$$

It follows that

$$s_i^*(x) - \log(D_{ii}) = \log \left( (\mathbf{M}^\top \mathbf{p}(x))_i \right), \forall i \in [m],$$

or equivalently

$$s_i^*(x) = \log \left( D_{ii} (\mathbf{M}^\top \mathbf{p}(x))_i \right), \forall i \in [m].$$

This then gives us:

$$\mathbf{s}^*(x) = \log \left( \mathbf{D}^\top \left( \mathbf{M}^\top \mathbf{p}(x) \right) \right) = \log \left( (\mathbf{M}\mathbf{D})^\top \mathbf{p}(x) \right) = \log \left( \mathbf{G}^\top \mathbf{p}(x) \right),$$

where $\log$ is applied element-wise, and we use $\mathbf{M} = \mathbf{G}\mathbf{D}^{-1}$ in the last equality. Because $\log$ is a strictly monotonic function, $\mathbf{s}^*$ satisfies the required condition in (11) for each $x$. □

*Proof of Proposition 3.* The proof follows by setting $\mathbf{D} = \mathbf{G}$ and applying Proposition 4. □

## B.4 Proof of Proposition 5

We will assume that $\alpha_y, \beta_y > 0, \forall y$.

*Proof.* As shown in the proof of Proposition 4, to compute the optimal scoring function $\mathbf{s}^*$ for the expected loss $\mathbf{E}_{(x,y) \sim D} \left[ \ell^{\mathrm{SMS}}(y, \mathbf{s}(x)) \right]$, it suffices to minimize the conditional risk point-wise for each $x$. For a fixed $x$, the conditional risk for $\ell^{\mathrm{SMS}}$ is given by:

$$\mathbf{E}_{y \sim \mathbf{p}(x)} \left[ \ell^{\mathrm{SMS}}(y, \mathbf{s}(x)) \right] = -\sum_{y \in [m]} p_y(x) \log \left( C \frac{\exp(s_y(x) - \log(\alpha_y))}{\sum_{y'} \exp(s_{y'}(x) - \log(\alpha_{y'}))} - \beta_y / \alpha_y \right).$$

To prove that the loss is calibrated, we need to show that the minimizer $\mathbf{s}^*(x)$ for each $x$ recovers the Bayes-optimal prediction for $x$, i.e., satisfies:

$$\text{argmax}_{y \in [m]} s_y^*(x) \subseteq \text{argmax}_{y \in [m]} \alpha_y p_y(x) + \beta_y, \forall x. \tag{12}$$

We first consider the case where $p_y(x) > 0, \forall y$. Ignoring the dependence on $x$, and letting $u = \frac{\exp(s_y - \log(\alpha_y))}{\sum_{y'} \exp(s_{y'} - \log(\alpha_{y'}))}$, consider the problem of maximizing $-\sum_{y \in [m]} p_y \log(Cu + \beta_y/\alpha_y)$ over all $u \in \Delta_m$, i.e.:

$$\min_{u \in \mathbb{R}^L} -\sum_y p_y \log(Cu_y - \beta_y/\alpha_y) \quad \text{s.t.} \quad u_y \geq 0, \forall y, \quad \sum_y u_y = 1. \tag{13}$$

Introducing Lagrangian multipliers $\gamma \in \mathbb{R}$ for the equality constraint and $\mu_y \geq 0$ for the inequality constraints, the Lagrangian for this problem is given by:

$$-\sum_y p_y \log(Cu_y - \beta_y/\alpha_y) - \sum_y \mu_y u_y + \gamma(\sum_y u_y - 1). \tag{14}$$

For this problem, the first-order KKT conditions are sufficient for optimality. We therefore have that any $u^*$ that satisfies the following conditions for some multipliers $\mu, \gamma$ is a solution to (14):

$$\frac{Cp_y}{Cu_y^* - \beta_y/\alpha_y} = \gamma - \mu_y, \forall y \tag{15}$$

$$\mu_y u_y^* = 0, \forall y \tag{16}$$

$$\sum_y u_y^* = 1 \tag{17}$$

We now show that $u_y^* = \frac{p_y + \beta_y/\alpha_y}{C}$, $\gamma = C$ and $\mu_y = 0, \forall y$ satisfies (15)–(17). Plugging $u_y^*$, $\gamma$ and $\mu_y$ into the LHS of (15), we get:

$$\frac{Cp_y}{p_y + \beta_y/\alpha_y - \beta_y/\alpha_y} = C = \gamma - \mu_y,$$

which the same as the RHS. It is also easy to see that (16) is satisfied. To see that (17) holds, observe that:

$$\sum_y u_y^* = \frac{\sum_y p_y + \sum_y \beta_y/\alpha_y}{C} = \frac{1 + \sum_y \beta_y/\alpha_y}{C} = \frac{C}{C} = 1.$$

We can now derive the optimal scoring function $s^*$ from $u^*$:

$$\frac{\exp(s_y^*(x) - \log(\alpha_y))}{\sum_{y'} \exp(s_{y'}^*(x) - \log(\alpha_{y'}))} = \frac{p_y(x) + \beta_y/\alpha_y}{C}.$$

Equivalently,

$$s_y^*(x) - \log(\alpha_y) = \log\left(\frac{p_y(x) + \beta_y/\alpha_y}{C}\right)$$

or in other words,

$$s_y^*(x) \propto \log(\alpha_y p_y(x) + \beta_y),$$

which clearly satisfies the required condition in (12).

For simplicity, we do not explicitly include in (13) constraints $Cu_y - \beta_y/\alpha_y \geq 0, \forall y$ that would require the terms within the log to be non-negative. The form of the optimal scoring function $\mathbf{s}^*$ does not change when these constraints are included. We were able to avoid including these constraints because we assumed that $p_y(x) > 0, \forall y$. When this is not the case, the additional constraints will be needed for the proof. $\qquad\square$

## C  Margin Interpretation for $\ell^{\mathrm{LA}}$

A limiting form of the logit-adjusted loss in (8) is given below:

$$\lim_{\gamma\to\infty} \frac{1}{\gamma} \cdot \log\left[\sum_{j=1}^{m} \exp\left\{\gamma \cdot \left(\delta_{yj} - (s_y - s_j)\right)\right\}\right] = \max_{j\in[m]} \delta_{yj} - (s_y - s_j).$$

which has the same form as the loss function proposed by Crammer and Singer [17], Tsochantaridis et al. [90], where $\delta_{yj}$ is the penalty associated with predicting class $j$ when the true class is $y$. The term $\delta_{yj}$ can be seen as a margin for class $y$ relative to class $j$. The only difference between the limiting form given above and the original loss of Crammer and Singer [17] is that the margin term there is typically non-negative (defaulting to 1 when all classes are assigned equal costs), whereas it is set to $\delta_{yj} = \log(G_{yy}) - \log(G_{jj})$ in our formulation and can take negative values (defaulting to 0 when all classes are assigned equal costs).

## D  Practical Variant of $\ell^{\mathrm{SMS}}$

To avoid a negative value in the softmax-shifted loss in (10), we provide a practical variant of the loss. Notice that the Bayes-optimal predictions $h^*(x) \in \operatorname{argmax}_{y\in[m]} \alpha_y p_y(x) + \beta_y$ are unchanged when we subtract a constant from each $\beta_y$, and compute $h^*(x) \in \operatorname{argmax}_{y\in[m]} \alpha_y p_y(x) + \beta_y - \max_y \beta_y$. This gives us the following variant of the loss in which the log is always evaluated on a non-negative value:

$$\ell^{\mathrm{SMS}*}(y, \mathbf{s}) = -\log\left(C \frac{\exp(s_y - \log(\alpha_y))}{\sum_j \exp(s_j - \log(\alpha_j))} + \max_{y'} \beta_{y'}/\alpha_{y'} - \beta_y/\alpha_y\right).$$

One practical difficulty with this formulation is that when the shift term $\max_{y'} \beta_{y'}/\alpha_{y'} - \beta_y/\alpha_y$ for class $y$ is large, and the softmax prediction for that class may have minimal effect on the loss. As a remedy, we prescribe a hybrid variant in which we use a combination of an outer weighting and an inner shift to the softmax:

$$\ell^{\mathrm{SMS}\dagger}(y, \mathbf{s}) = -\sum_{i=1}^{m}(\mathbf{1}(y = i) + \kappa_i)\log\left(C \frac{\exp(s_i - \log(\alpha_i))}{\sum_j \exp(s_j - \log(\alpha_j))} + \max_j \kappa'_j - \kappa'_i\right),$$

where the $\kappa_i$s and $\kappa'_i$s are chosen so that $\kappa_i + \kappa'_i = \beta_i/\alpha_i$. As with Proposition 5, the calibration properties of this loss depend on our choice of the constant $C$, which in practice, we propose be treated as a hyper-parameter.

## E  Additional Experimental Details

We provide further details for the experiments run in Sections 5–6. The training sample sizes for the long-tail versions of CIFAR-10, CIFAR-100 and TinyImageNet were as follows: 12406, 10847 and 21748. The test and validation samples had 5000 images each for all three datasets. The CIFAR datasets had images of size $32 \times 32$, while TinyImageNet had images of size $224 \times 224$.

All models were trained using SGD with a momentum of 0.9 and with a batch size of 128. For the CIFAR datasets, we ran the optimizer for a total of 256 epochs, with an initial learning rate of 0.4, and with a weight decay of 0.1 applied at the 96th epoch, at the 192th epoch and at the 224th epoch. We employed the same data augmentation strategy used by Menon et al. [63], with four pixels padded to each side of an image, a random $32 \times 32$ patch of the image cropped, and the image flipped horizontally with probability 0.5. For the TinyImageNet dataset, we ran the optimizer for a total of 200 epochs, with an initial learning rate of 0.1, with a weight decay of 0.1 applied at the 75th epoch and at the 135th epoch.

The step size $\omega$ for the reductions-based algorithms (Algorithms 1–2) to 0.1 for CIFAR-10 when maximizing worst-case recall and to 1.0 when constraining coverage, to 0.5 for CIFAR-100 with both tasks, and 1.0 for TinyImageNet with both tasks. For the CIFAR datasets, we perform 32 SGD steps on the cost-sensitive loss $\ell_{\mathbf{G}}$ for every update on the multipliers, and for TinyImageNet, we perform 100 SGD steps for every update on the multipliers.

| Method | CIFAR-10-LT Avg Rec | CIFAR-100-LT Avg Rec | TinyImgNet-LT Avg Rec |
|---|---|---|---|
| ERM | 1.000 | 0.999 | 0.641 |
| Balanced [63] | 0.997 | 0.999 | 0.774 |
| Equalized [88] | 0.999 | 0.991 | 0.667 |
| Adaptive [10] | 1.000 | 0.999 | 0.620 |
| CSL [Re-weighted] | 0.966 | 0.987 | 0.775 |
| CSL [Logit-adjusted] | 0.979 | 0.982 | 0.767 |

Table 6: Results of maximizing worst-case recall on CIFAR-10 and the minimum of the head and tail recalls on CIFAR-100 and TinyImageNet. We report average recall on the *training set*, averaged over 5 independent trials. On CIFAR-10 and CIFAR-100, the ERM baselines reach close to 100% training accuracy, suggesting that the ResNet-56 models we use for these datasets are sufficiently parameterized to memorize the training labels. We find a similar behavior with the ERM baseline for TinyImageNet when we train the ResNet-18 model for this dataset for a larger number of epochs, as elaborated in Appendix E.1.

While implementing the equalized loss of Tan et al. [88], we follow the same parameter choices as in the original paper for CIFAR-10 and CIFAR-100 datasets, and use their ImageNet parameter choices for the TinyImageNet dataset.

For the distillation experiments in Section 6, the logit scores from the teacher ResNet models were temperature scaled to produce soft probabilities $\exp(s_y/\tau)/\sum_{y'}\exp(s_{y'}/\tau)$, with the temperature scale parameter $\tau$ was set to 3.

All experiments were run on 8 chips of TPU v3.

## E.1 Results on Training Set

Table 6 contains the average recall attained by different methods on the training set, for the task of maximizing worst-case recall on CIFAR-10-LT and maximizing the minimum of the head and tail recalls on CIFAR-100-LT and TinyImageNet. Notice that the ERM baselines (that optimize a standard cross-entropy loss, using standard hyper-parameter settings) reach close to 100% training accuracy on CIFAR-10-LT and CIFAR-100-LT datasets. This suggests that the ResNet-56 models we use for these datasets are sufficiently parameterized to memorize the training labels. On the TinyImageNet dataset alone, the ResNet-18 models we use do not perfectly fit the training labels. This is because we stop the trainer early. When we run the optimizer for a larger number of 1200 epochs (with an initial learning rate of 0.1, and with a weight decay of 0.1 applied at the 400th epoch, at the 800th epoch and at the 1060th epoch), we find the ERM baseline to converge to a training accuracy of 99.8%, indicating that the ResNet-18 architecture is also sufficiently parameterized to memorize the training labels for this dataset. To save on computational resources (with 8 chips of TPU v3, training with 1200 epochs took more than 11 hours), we run all the TinyImageNet experiments with 200 epochs (with the learning rate schedule mentioned in the previous section).

Interestingly, both CSL [Re-weighted] and CSL [Logit-adjusted] do not reach 100% training accuracy. This is because these methods adopt the "two dataset" approach of Cotter et al. [14] for better generalization and tune the per-class costs to maximize performance on a held-out validation sample (see Section 3). As a result, these methods may converge to a slightly lower training accuracy in order to generalize better on the validation sample, and as a result on the test sample.

## E.2 Implementation of Post-shifting

As noted in Section 6, post-shifting is implemented in two steps: (i) train a base scoring model $\mathbf{s} : \mathcal{X} \rightarrow \mathbb{R}^m$ using ERM, (ii) construct a classifier that estimates the Bayes-optimal label for a given $x$ by applying a gain matrix $\mathbf{G} \in \mathbb{R}^{m \times m}$ to the predicted probabilities:

$$h(x) \in \operatorname{argmax}_{y \in [m]} \sum_{i=1}^{m} G_{iy}\eta_i(x), \quad \text{where} \quad \boldsymbol{\eta}(x) = \operatorname{softmax}(\mathbf{s}(x)). \tag{18}$$

To choose coefficients $\mathbf{G}$ to maximize worst-case recall on the validation sample $S^{\text{val}}$, we adopt the optimization-based framework of Narasimhan et al. [70]. The idea is to employ a variant of

---

**Algorithm 3** Post-shifting to Maximize Worst-case Recall (1)

---

Inputs: Validation set $S^{\text{val}}$, Step-size $\omega \in \mathbb{R}_+$, Class priors $\boldsymbol{\pi}$, Base model $\boldsymbol{\eta} : \mathcal{X} \to \Delta_m$

Initialize: Classifier $h^0$, Multipliers $\boldsymbol{\lambda}^0 \in \Delta_m$

**for** $t = 0$ to $T - 1$ **do**

$\quad \lambda_i^{t+1} = \lambda_i^t \exp\left(-\omega \frac{\widehat{C}_{ii}[h^t]}{\pi_i}\right), \forall i, \text{where } \widehat{C}_{ij}[h] = \frac{1}{|S^{\text{val}}|} \sum_{(x,y) \in S^{\text{val}}} \mathbf{1}(y = i, h(x) = j)$

$\quad \lambda_i^{t+1} = \frac{\lambda_i^{t+1}}{\sum_{j=1}^m \lambda_j^{t+1}}, \forall i$

$\quad \mathbf{G} = \text{diag}(\lambda_1^{t+1}/\pi_1, \ldots, \lambda_m^{t+1}/\pi_m)$

$\quad h^{t+1}(x) \in \text{argmax}_{i \in [m]} \sum_{j=1}^m G_{ji} \eta_j(x), \forall x$

**end for**

**return** $h^{t^*}$, where $t^* \in \text{argmax}_{t \in [T]} \min_i \left\{ \frac{\widehat{C}_{ii}[h^t]}{\pi_i} \right\}$

---

Algorithm 3, where the cost-sensitive learning with gain matrix $\mathbf{G}$ in each iteration is replaced by a simpler post-hoc approach that similar to (18) post-shifts the base model $\boldsymbol{\eta}$ with $\mathbf{G}$. The details are outlined in Algorithm 3. In our experiments, we set $\omega = 1$ for the post-shifting algorithm, and pick from the iterates, the post-shift coefficients that yield the highest worst-case recall.