# OpenReview forum: "Training Over-parameterized Models with Non-decomposable Objectives"
_NeurIPS.cc/2021/Conference — NeurIPS 2021 Poster_

### Official Review · Reviewer_t7Bj · 2021-07-14

**Rating:** 7
**Confidence:** 2

**Summary:**

The paper considers multiclass classification with "non-decomposeable" objectives: that is, objectives that are not the sum of individual terms over sample points. The focus is on problems that can be reformulated in terms of a single saddle point structure. These are often recast as a sequence of cost-sensitive problems, but this strategy can be bad when the learning method applied at each step tends to achieve zero error. The authors show that considering instead a sequence of problems with non-standard losses losses leads to significantly better empirical results. A partial justification for the non-standard losses is that they are calibrated (so their optima correspond to optimal classifiers).

**Ethical Concerns:**

None.

**Limitations And Societal Impact:**

They have.

**Main Review:**

In a classification problem with $m$ classes, each classifier $h:\mathcal{X}\to [m]$ has an associated confusion matrix $C[h]$ that describes how often it predicts a $j$ when the true label is a $i$. The "non-decomposeable" objectives described in the title of the paper are losses that can be expressed in a saddle point fashion in terms of $C[h]$, like for example:
$$(\star) \max_{h\in mathcal{H}} \min_{\lambda\in S}\sum_{i,j}G_{ij}(\lambda)C_{ij}[h] ,$$
where the matrix $G$ depends linearly on $\lambda$. Such metrics are quite natural and include e.g. the minimum recall over classes for a given $h$.

The present submission notices that, for a fixed $\lambda$, $(\star)$ is a cost-sensitive classification problem that can be solved by standard techniques. Therefore, a solution to the saddle point problem $(\star)$ could obtained via a sequence of these cost-sensitive problems alternated with updates to $\lambda$. However, this strategy may perform poorly for finite samples. Intuitively, when the base learner tends to interpolate data, then the cost-sensitivity constraint is irrelevant, as training loss will be zero anyway.

This paper presents an alternative approach whereby the cost-sensitive learning problems are replaced by certain learning problems. The salient feature of the latter is that the losses used are calibrated: that is, a classifier that achieves minimum population surrogate loss is the optimal one corresponding to an "ideal" iteration of the saddle point problem. Calibration is not quite a practical result, since in practice it is rarely the case that the population surrogate loss is exactly minimized. Still, this property lends some credibility to the procedure developed here. Moreover, the method performs well in experiments with CIFAR-10, CIFAR-100 and TinyImageNet. The authors also consider the effect of post shifting and distillation on the final result.

This submission is quite well written. Mathematical statements and proofs are mostly well explained, and the authors give very precise descriptions of their experiments. As I explained above, calibration is not a decisive result proving their approach works, but it is nonetheless an interesting result. To this reader, the experimental results are convincing ; however, I would have liked to have access to code, and larger experiments would make a more compelling case. Overall, my evaluation of the paper is positive, although I also feel I am not a specialist on the topics of the submission.

An additional (minor) comment is that the proof of Prop. 2 does not seem to work if there are repeated feature vectors with distinct labels in the dataset $S$.








**Time Spent Reviewing:**

4h

---

> ### Author Response · Authors · 2021-08-06
> **Response to Reviewer t7Bj: We'll make the code publicly available**
>
> Thank you for the thorough feedback and positive comments.
>
> **Calibration**: We agree that the calibration results largely serve as sanity checks for our approach, and it is the experimental results that show the significant advantage that our proposal has over existing baselines.
>
> **Code**: We’ll definitely make the code publicly available if the paper gets accepted.
>
> **Prop. 2**: Thanks for catching this. We’ll add an assumption that the instances $x$ in the training sample $S$ are all unique.

---

### Official Review · Reviewer_svfi · 2021-07-16

**Rating:** 6
**Confidence:** 4

**Summary:**

The author proposed a new approach for optimizing non-decomposable metrics that focuses on over-parameterized models (i.e., deep learning models). Rather than performing reduction and reweighting as in the traditional models for non-decomposable metrics, the proposed model modifies the logits of the over-parameterized models. The author also shows that the logit modifications are calibrated with respect to the original metric. Finally, the author shows the effectiveness of the proposed models on CIFAR and TIN datasets.


**Limitations And Societal Impact:**

Yes

**Main Review:**

Overall, I like the idea of modifying the logits instead of reweighting the samples or classes, as it has been shown in previous literature that sample reweighting does not work as intended in the over-parameterized models. The author shows that the proposed logit modification can align the over-parameterized models with the original metrics better. I also like the consistency property of the proposed loss.
I have questions that I would like the author to address.
1. Since the author mentioned that the proposed method works for complex metrics, could the author explain how generic the model is. Could the model handle other types of non-decomposable metrics? For example, the F-score or the MCC score, or ranking-based scores. If someone wants to apply the proposed model on a particular non-decomposable metric, how much modification is needed?
2. Related to the first question, how to construct the G matrix for a particular metric?
3. The logit adjustment technique presented in the paper is the inner step of the full training algorithm, but it was not so clear from the main paper. I suggest the author provide more explanation in the main paper rather than putting it in the appendix.
4. Some related papers are missing from the reference: e.g., Fathony & Kolter (2020) and Luo et. al. (2021)

Fathony, R. & Kolter, Z.. (2020). AP-Perf: Incorporating Generic Performance Metrics in Differentiable Learning. Proceedings of the Twenty Third International Conference on Artificial Intelligence and Statistics, in PMLR 108:4130-4140

Luo, Junru, Hong Qiao, and Bo Zhang. "A Minimax Probability Machine for Non-Decomposable Performance Measures." arXiv preprint arXiv:2103.00396 (2021).

**Time Spent Reviewing:**

4

---

> ### Author Response · Authors · 2021-08-06
> **Response to Reviewer svfi: Our approach extends to general metrics such as F-score**
>
> Thank you for the positive and encouraging comments.
>
> (1) **Extension to general metrics**: This is a great question. Yes, the proposed approach does extend to metrics such as the F-score (by introducing auxiliary variables), as well as AUC-based metrics (via a Riemann approximation to the integral). Ref. [25] provides details of how the optimization of these metrics can be posed as constrained optimization problems, and in turn as cost-sensitive learning tasks. More generally, using the reduction techniques from ref. [68], any learning problem of the following form can be reduced to cost-sensitive learning problems, and thus tackled by the proposed approach:
>
> $~~~~~~\max_{h} \psi(C[h])   ~~~~s.t.~~~~  \phi_k(C[h]) \leq 0, ~~k = 1, …, K,$
>
> where $C[h]$ is the confusion matrix for classifier $h$, and $\psi$ and $\phi_k$s are *convex* (or *fractional-linear*) functions of $C$. Their reduction techniques are generic and only require to be able to compute gradients for $\psi$ and $\phi_k$s.
>
> (2) **$G$-matrix for general metrics**: Similar to the formulation in Section 2, the $G$-matrix for general metrics of the above form can be constructed by writing out the Lagrangian for the problem, and using the Lagrange multipliers to determine the weights $G_{ij}$ on individual classes. Ref. [68] provides details of the Lagrangian primal-dual optimization for different metrics. For example, to maximize the F-score: $\frac{2TP}{2TP + FP + FN}$, we would introduce slack variables $\xi_1$ and $\xi_2$ for the numerator and the denominator, and reformulate the problem as:
>
> $~~~~~~\max_{\xi_1, \xi_2 \in [0, 1]} \frac{\xi_1}{\xi_2} ~~~~s.t.~~~~ \xi_1 \leq 2TP, ~~~\xi_2 \geq 2TP + FP +FN.$
>
> The Lagrange multipliers $\lambda_1$ and $\lambda_2$ for the constraints can then be used to derive weights on TP, FP and FN, (i.e., the $G$ matrix). We'll be happy to include more details about the derivation of $G$ in the paper.
>
> (3) **Detailed algorithm**: Thanks for the suggestion. We’ll move details of the algorithms from Appendix A to the main text (by perhaps cutting down on Section 6).
>
> (4) Thanks for the additional references. We’ll be sure to cite them.
>
> **References:**
>
> &emsp;[25] Elad et al., "Scalable learning of non-decomposable objectives", AISTATS 2017.
>
> &emsp;[68] Narasimhan et al., "Optimizing generalized rate metrics with three players", NeurIPS 2019.

---

> > ### Comment · Reviewer_svfi · 2021-08-29
> > **Thanks for the response**
> >
> > Thanks the authors for addressing my questions and concerns.
> > I would suggest the author to provide more clarity to the reader as well as explanation on how to use the method for more general metrics as it is important to the reader that may want to apply it to a slightly different problem.

---

### Official Review · Reviewer_e3jE · 2021-07-16

**Rating:** 6
**Confidence:** 3

**Summary:**

The paper proposes to use logit adjustments for class imbalanced training settings with over-parameterized models.
They introduce logit adjustments with general gain matrices (in constrast to only diagonal gain matrices).


**Limitations And Societal Impact:**

The empirical evaluation lacks baselines and the scope of contribution is limited.

**Main Review:**

The paper is well written, and nicely introduced.
The work is very interesting, however, it seems like the contribution of the work is limited considering the related works.

There are two very similar works (which have been cited by the paper). https://arxiv.org/pdf/2007.10740.pdf https://arxiv.org/pdf/2007.07314.pdf
Ren et al. (https://arxiv.org/pdf/2007.10740.pdf) also use logit adjustment on the same task, while they use a ResNet-34 instead of the here used ResNet-56.
It is not clear to me, why ResNet-56 is much more overparameterized than ResNet-34. I see that it is larger, but it is not that much larger.
The operation on overparameterized models seems to be the main contribution of this paper, it does not convince me that the proposed setting is overparameterized and (that at the same time) in the work by Ren et al. it is not overparameterized.

I acknowledge that this work is a generalization of those previous works.
However, the empirical distinction of your work to the previous works is not strong.
As Menon et al. and Ren et al. train on the same task, I would strongly encourage you to include these baselines (as well as some of their baselines) into your paper.

More clearly distinguishing between prior work and your contributions and using prior works as baselines would improve my score.

Minor:
l. 143 function -> functions

Comment on Confidence:
I carefully checked the math in the main paper.

**Time Spent Reviewing:**

4.25

---

> ### Author Response · Authors · 2021-08-06
> **Response to Reviewer e3jE: We already compare to Menon et al.**
>
> Thank you for reviewing our paper and providing a detailed feedback.
>
> **Comparison to Menon et al. already in Tables 2 & 3**:
> The reviewer asks for a comparison to the work of Menon et al. (2021) and Ren et al. (2020). We wish to emphasise four points.
>
> - First, we *do already compare* with the class-prior adjusted loss of Menon et al. (please see results for “LA with class priors”). Ren et al. also propose a similar logit-adjusted loss based on class frequencies, along with an additional meta-sampling strategy.
>
> - Second, *conceptually*, a key difference with both these methods is that they seek to optimize “balanced accuracy” (a.k.a. average recall in our tables), which is *fundamentally* different from the more complex “worst-case recall” metric or the “recall-coverage” metric we seek to optimize in our paper. Unlike balanced accuracy, the **metrics we handle are non-decomposable**, i.e. cannot be expressed as an average of per-example errors, and require solving more complicated min-max optimization problems. To the best of our knowledge, ours is the first paper to handle training of *over-parameterized* models with such a *general family of multiclass classification metrics* (which also includes F-measure and other common metrics; L103–104).
>
> - Third, *empirically*, **our experiments confirm the inadequacy of simply optimizing for balanced accuracy (e.g., using the method of Menon et al.)**, when the goal is to optimize some more complex metric. For example, in the results in Tables 2, the loss proposed by Menon et al. performs the best on average recall/balanced accuracy (a metric they optimize), but performs poorly on the worst-case (min) recall metric that we care about. Given that our optimization goal is very different from Menon et al. and Ren et al., we do not expect the baselines they compare with to fare substantially better on the complex metrics we consider. For example, **we ran some of the important baselines from Menon et al.** for the min-recall maximization task in Table 2, and found that our approach yields significant gains over those methods on min recall (the additional baselines are competitive on average recall, but perform poorly on the tail classes):
> \\begin{array}{|c|c|c|c|c|c|c|c|c|}
> \\hline
>     & \text{CIFAR10-LT} & \text{CIFAR100-LT}  & \text{TinyImageNet-LT} \\\\ \\hline\\hline
>     & \text{Min Rec} & \text{Min Rec} & \text{Min Rec}   \\\\ \\hline\\hline
> \text{LA with class priors (Menon et al., '21)} &   0.658  &   0.200  &  0.095 \\\\ \\hline
> \text{Equalised (Tan et al., '20)}  &  0.637 & 0.095  &  0.020 \\\\ \\hline
> \text{Adaptive (Cao et al., '19)} & 0.538 & 0.112  &  0.002 \\\\ \\hline
> \text{Ours} & {\bf 0.721} & {\bf 0.418} & {\bf 0.310} \\\\ \\hline
> \\end{array}
>
> - Fourth, **we do compare against the state-of-the-art methods for optimizing non-decomposable metrics, including Cotter et al. [13] (CSL [re-weighted]) and post-shift strategies [86] (Section 6)**, which we believe are the most relevant baselines for our experimental settings.
>
> We’ll definitely include a more detailed discussion on the differences with Menon et al. and Ren et al., and on our experimental comparisons to Menon et al. in Tables 2–3. We'll also be happy to include the additional comparisons to baselines in Menon et al.

---

> > ### Author Response · Authors · 2021-08-24
> > **Follow-up: Response to Reviewer e3jE**
> >
> > Dear Reviewer e3jE,
> >
> > Thanks again for taking the time to provide a detailed review. Please let us know if there are any remaining questions that you would like us to clarify following our response.

---

> > > ### Comment · Reviewer_e3jE · 2021-08-24
> > > **Response**
> > >
> > > Thank you for answering most of my concerns.
> > >
> > > I understand that a core aspect of your work is "over-parameterized models"; however, it is still not clear to me how you approach this empirically.
> > > You did not answer to "It is not clear to me, why ResNet-56 is much more overparameterized than ResNet-34. I see that it is larger, but it is not that much larger. The operation on overparameterized models seems to be the main contribution of this paper, it does not convince me that the proposed setting is overparameterized".
> > > I just now realize that my statement was not correct because ResNet-34 has 21 million parameters while ResNet-56 has only 0.85 million parameters.
> > > Under that consideration, I would not consider ResNet-56 to be overparameterized.
> > > Why do you consider it to be overparameterized?
> > >
> > > Do you consider deep learning to be overparameterized? If so, potentially the focus should not be on overparameterization but instead on deep learning?
> > > Do you consider overparameterized to mean too many parameters for the data set, such as ResNet-101 on MNIST?
> > >
> > > Overall, I am willing to change my score.

---

> > > > ### Author Response · Authors · 2021-08-25
> > > > **Definition of over-parameterization**
> > > >
> > > > Dear Reviewer e3jE,
> > > >
> > > > We’re sorry that we missed your earlier question on over-parameterization. We consider a model to be “over-parameterized” if it **contains enough parameters to perfectly fit the training data**. We believe that this definition of “over-parameterization” is fairly common in the literature [A, B]. Indeed such a condition holds with the state-of-the-art deep learning models such as ResNet, where the number of learnable parameters in these models is often much larger than the number of the training samples. For example, the ResNet-56 architecture we use for the CIFAR-100-LT dataset contains 552K parameters, which is far higher than the size of the training set (~11K), and we confirm that the networks we train perfectly fit the training labels.
> > > >
> > > > That said, we would like to clarify that the main distinction between our paper and the prior work of Menon et al. and Ren et al. is **not in the number of model parameters** we can handle (in fact we adopt the same architectures as in Menon et al.), but in the **wider range of training objectives** that we can handle. Indeed, both Menon et al. and Ren et al. work with over-parameterized ResNet networks, but seek to optimize the “balanced accuracy” metric. Our focus, on the other hand, is on training over-parameterized models to optimize a wider range of “complex” (non-decomposable) evaluation metrics. Per our results in Sec 5, and comments in the response above, naively using the methods of Menon et al. and Ren et al. can lead to significant degradation with respect to such “complex” metrics. We’ll be happy to clarify this in the main paper.
> > > >
> > > > We would ideally prefer to not broaden the scope of the paper to cover “general deep learning models”, as we are specifically interested in scenarios where the model is trained to perfectly fit the training set and where it’s unclear how the model will generalize on a held-out test set (for a given evaluation metric of interest).
> > > >
> > > > Additional refs:
> > > >
> > > > [A] Arora et al. “Fine-Grained Analysis of Optimization and Generalization for Overparameterized Two-Layer Neural Networks”, ICML 2019.
> > > >
> > > > [B] Zhang et al., “Understanding deep learning requires rethinking generalization”, ICLR 2017.
> > > >
> > > > Please let us know if you need further clarifications.

---

> > > > > ### Comment · Reviewer_e3jE · 2021-08-25
> > > > > **Response**
> > > > >
> > > > > Thank you for the quick response.
> > > > >
> > > > > I find it surprising that a model with 552k parameters perfectly fits the training data of 11k examples, which are each high dimensional. Do I understand correctly that all of your trained models have 100% training accuracy, right? Is it only that they could reach 100%, or do you always train them to 100% accuracy?
> > > > > In that case, I agree that the model is overparameterized. From experience, I had not expected that the training accuracy would reach 100% for the described settings.

---

> > > > > > ### Author Response · Authors · 2021-08-25
> > > > > > **Re: Response**
> > > > > >
> > > > > > Firstly, we would like to thank you for pushing us to be more precise about our definition of “over-parameterization”.
> > > > > >
> > > > > > > Do I understand correctly that all of your trained models have 100% training accuracy, right? Is it only that they could reach 100%, or do you always train them to 100% accuracy?
> > > > > >
> > > > > > Our ResNet models have the *capacity* to reach 100% training accuracy; further, when trained with SGD and standard hyper-parameter settings, they do indeed achieve 100% accuracy.
> > > > > >
> > > > > > For the first point, in the paper, we confirm this in the illustrative experiments we showcase in Figure 1, where we compare ResNet-56 trained on 10K images from CIFAR-10 on three different losses and mention that the models plotted are trained to reach near-zero training error (most models over the 5 repeated runs achieve 100% training accuracy, with the lowest accuracy being 99.93%). Note that the ability of ResNet models to achieve perfect training error on CIFAR has been observed before, e.g., He et al., "Deep residual learning for image recognition", CVPR 2016 (Figure 6).
> > > > > >
> > > > > > For the second point, in our experiments in Section 5, we can confirm that the ERM baseline models (that optimize a standard cross-entropy loss, using standard hyper-parameter settings) do reach 100% training accuracy on e.g. the CIFAR-100-LT dataset. Our proposed “CSL [Logit-adjusted]” for this experiment reaches a training accuracy of 98.4%. The reason for the slightly lower training accuracy is that our method adopts the “two dataset” approach of Cotter et al. for better generalization and tunes the per-class costs to maximize performance on a held-out validation sample (more details in L184-193 and L250-254). As a result, our method may converge to a slightly lower training accuracy in order to generalize better on the validation sample. Note that all the numbers reported in the paper are on a test sample different from the validation sample.
> > > > > >
> > > > > > > I find it surprising that a model with 552k parameters perfectly fits the training data of 11k examples, which are each high dimensional.
> > > > > >
> > > > > > Our intuition is that with ~50x more parameters than examples, it should be possible for the model to perfectly fit the training data. Even if the examples are high-dimensional to begin with, the model can always use a small set of informative features (or a lower-dimensional embedding of the images) to identify a unique feature combination for each training example, and memorize the training label for that combination.
> > > > > >
> > > > > > We’ll be happy to report the training accuracies for all experiments (perhaps in the appendix), and have a detailed discussion in the main text on what we mean by “over-parameterization” and the implications that it has on the training performance of the different methods.
> > > > > >
> > > > > > Please feel free to let us know if you need further clarification.

---

> > > > > > > ### Comment · Reviewer_e3jE · 2021-08-25
> > > > > > > **Thanks**
> > > > > > >
> > > > > > > Thank you for the clarifications.
> > > > > > > Adding this information to the paper will improve it.
> > > > > > > Accordingly, I increase my score by 1.

---

### Official Review · Reviewer_qkBf · 2021-07-18

**Rating:** 6
**Confidence:** 2

**Summary:**

This paper proposed new cost-sensitive losses for training over-parameterized models. The losses are "calibrated" (see details in the main review). The experiments show the proposed method has gained in terms of a certain metric.

**Limitations And Societal Impact:**

The clarity of the paper can be improved. See details above.

**Main Review:**

First of all, I have to claim that the paper is beyond my expertise and I'm not familiar with the literature. I asked to have the paper unassigned but it didn't work out. I'm not confident about my review.

Clarity & Significance:

The paper is not so clear for me (perhaps it is just because I'm not an expert).

I'm not quite familiar with the terminology "calibrated", does it mean "Fisher consistency" or something related? I mean, optimizing a certain surrogate loss can achieve the Bayes optimal classifier given infinite samples with a hypothesis of all measurable functions. It should be defined rigorously.

The paper also presents some propositions. Only proposition 1 is quoted from existing work. I'm not quite sure about the others. Any of them is the main theory of this paper? They are the same and natural/simple to me according to the presentation. If there is some technical contribution, please highlight it.

Further, the discussion of the literature, especially the related ones is not sufficient. For instance, the authors claim that Eqn. 7 is a simple generalization of existing work. But in which sense? Making them "calibrated" or something else? I didn't see the significance and necessity.

As for the experiments, the results of the methods differ a lot under two different measures. The proposed method does not perform as well as the baseline in terms of the avg. acc. Then which one do we really care about and any deeper analysis? Lastly, the experimental settings are typical? It seems quite arbitrary to define a classification task. It would be better to consider some widely adopted settings and compare to a larger family of existing methods.


Originality & Quality:

I cannot tell the originality of the paper as I do not know the literature at all. I give a temporal weak reject and update my score based on the authors' feedback and discussion with other reviewers.



**Time Spent Reviewing:**

2

---

> ### Author Response · Authors · 2021-08-06
> **Response to Reviewer qkBf: Clarification on novelty & significance, and other comments**
>
> Thank you for reviewing our paper and providing detailed feedback. Below are our responses for the main comments.
>
> **Main contribution**: Our main contribution (L31) is a principled approach for training over-parameterized models to optimize complex evaluation metrics.
>
> The *novelty* of this contribution compared to prior works such as Menon et al. (2021) is that the latter focused on training over-parameterized models with simple metrics (like classification accuracy or average recall), while we handle a wider family of complex evaluation metrics (see Table 1) popular in practice. Importantly, unlike the prior work, the metrics we optimize are “non-decomposable” (i.e., cannot be expressed as an average of per-example errors), and can depend on both the diagonal and off-diagonal entries of the confusion matrix.
>
> The *significance* of this contribution is illustrated by our experiments (Table 2-3), which confirm that simply using methods such as Menon et al. can lead to models that vastly underperform on complex metrics (e.g., worst-class recall) compared to the proposed method. Such complex metrics arise in several real-world applications, such as fairness.
>
> **Classification calibration**: The reviewer is correct that this is the same as Fisher consistency. Loosely speaking, a surrogate loss $\ell: [m] \times \mathbb{R}^m \rightarrow \mathbb{R}_+$ is classification calibrated for a metric if the minimizers of the conditional risk:
>
> $\mathbb{E}_{y|x}\left[\ell(y, \mathbf{s}(x))\right]$
>
> result in Bayes-optimal classifiers for the metric. Such terminology is fairly established in the statistical learning literature (e.g., Bartlett et al. "Convexity, classification, and risk bounds," 2006). However, we agree it would be useful to include for completeness, and are happy to do so.
>
> **Significance of theoretical results**: Propositions 3–5 show classification calibration for the different losses proposed in the paper (proofs provided in Appendix B). These results confirm that our methods do work in the large sample limit, and lend credibility to our proposal.
>
> **Clarification on Eq. 7**: Menon et al. analyze a loss similar to Eq. 7, with the difference being that they set the per-class weights $G_{yy}$ to the inverse class priors. Hence we refer to Eq. 7 as a “simple generalization” of their loss. We also note that when $G$ is non-diagonal, this simple modification will not work, and so we provide new calibrated losses for the general case of non-diagonal weight matrices $G$ (see Eqs. 9–10).
>
> **Are the experimental settings typical?** The settings we consider do arise in many real-world applications. For example, several recent papers on fairness have highlighted the importance of measuring the “worst-case” accuracy across different groups (e.g., [18, 11]), and it is natural that this is a metric we may want to optimize while training a classifier. Similarly, constraining for a specific coverage or precision target is a standard requirement in many production ML systems, with several libraries and prior works providing optimization strategies for such metrics (e.g., [2, 3, 15, 25, 64]). We do compare against the state-of-the-art baselines for these problems, including the constrained optimization approach of Cotter et al. [13] (CSL [re-weighted]), the balanced loss of Menon et al. [60] (LA with class priors), and post-hoc strategies [86] (Section 6).
>
> **Which evaluation metric we care about?**: In Table 2, the metric we care about is the *worst-case (min) recall*, and our approach significantly beats all the baselines on this metric. We also report the average recall, as this is the metric optimized by the “LA with class priors” baseline of Menon et al. Unsurprisingly, while the baseline performs the best on average recall, it fares poorly on min recall, which is the main metric of evaluation. Trading off performance on one metric to do well on another is fairly common in the literature, and is often unavoidable as the form of the Bayes classifier can be very different for the two metrics (see e.g. Koyejo et al. [44]). In Table 3, our goal is to solve a constrained optimization problem: *maximize average recall subject to a constraint on the classifier’s “coverage”*. Among all methods that satisfy the coverage constraint, our approach performs the best on average recall. Again, the “LA with class priors” baseline is the best on average recall, but notably, does not satisfy the coverage constraint.

---

> > ### Comment · Reviewer_qkBf · 2021-08-19
> > **Thanks for clarification**
> >
> > I appreciate the authors for their detailed feedback. I realized that most of my concerns are misunderstandings and they are clarified. Therefore, I would like to raise my score.

---

> > > ### Author Response · Authors · 2021-08-21
> > > **Thanks to the reviewer for revising their score!**
> > >
> > > We really appreciate the reviewer for going through our response and revising their score. Thank you!

---

### Decision · Program_Chairs · 2021-09-27

**Decision:**

Accept (Poster)

**Comment:**

The paper proposes new calibrated cost-sensitive losses for training over-parameterized models.  The reviewers were unanimous in their opinion that the paper is above the acceptance threshold.  The approach is well supported theoretically, and covers an interesting family of non-decomposable losses.  The authors have promised to release code at the time of publication in response to a reviewer comment.